# Direct Bias-Correction Term Estimation for Average Treatment Effect Estimation

## Abstract

This study considers the estimation of the direct bias-correction term for estimating the average treatment effect (ATE). Let $\{(X_i, D_i, Y_i)\}_{i=1}^n$ be the observations, where $X_i \in \mathbb{R}^K$ denotes $K$-dimensional covariates, $D_i \in \{0, 1\}$ denotes a binary treatment assignment indicator, and $Y_i \in \mathbb{R}$ denotes an outcome. In ATE estimation, $h_0(D_i, X_i) := \frac{\mathbb{1}[D_i=1]}{e_0(X_i)} - \frac{\mathbb{1}[D_i=0]}{1-e_0(X_i)}$ is called the bias-correction term, where $e_0(X_i)$ is the propensity score. The bias-correction term is also referred to as the Riesz representer or clever covariates, depending on the literature, and plays an important role in construction of efficient ATE estimators. In this study, we propose estimating $h_0$ by directly minimizing the Bregman divergence between its model and $h_0$, which includes squared error and Kullback–Leibler divergence as special cases. Our proposed method is inspired by direct density ratio estimation methods and generalizes existing bias-correction term estimation methods, such as covariate balancing weights, Riesz regression, and nearest neighbor matching. Importantly, under specific choices of bias-correction term models and Bregman divergence, we can automatically ensure the covariate balancing property. Thus, our study provides a practical modeling and estimation approach through a generalization of existing methods.

## 1 Introduction

We consider the problem of estimating the average treatment effect (ATE) in causal inference (Imbens & Rubin, 2015). Methods for estimating ATEs are typically designed to eliminate bias arising from treatment assignment and the estimation of nuisance parameters, aiming for (asymptotic) unbiasedness and efficiency.

### 1.1 ATE estimators and bias correction

We begin by formulating the problem. There are two treatments, denoted by 1 and 0.[1] For each treatment $d \in \{1, 0\}$, let $Y(d) \in \mathbb{R}$ denote the potential outcome under treatment $d$. The treatment assignment indicator is denoted by $D \in \{1, 0\}$, and the observed outcome is given by $Y = \mathbb{1}[D = 1]Y(1) + \mathbb{1}[D = 0]Y(0)$, meaning that we observe $Y(d)$ only if the unit is actually assigned to treatment $d$. Each unit is characterized by $K$-dimensional covariates $X \in \mathcal{X} \subset \mathbb{R}^K$, where $\mathcal{X}$ denotes the covariate space. For $n$ units indexed by $1, 2, \ldots, n$, let $\mathcal{D} := \{(X_i, D_i, Y_i)\}_{i=1}^n$ denote the observed data, where each $(X_i, D_i, Y_i)$ is an i.i.d. copy of $(X, D, Y)$ generated from an underlying distribution $P_0$. Our goal is to estimate the ATE, defined as

$$\tau_0 := \mathbb{E}\big[Y(1) - Y(0)\big],$$

where the expectation is taken over the distribution $P_0$. Note that we can also apply our method for the ATE for the treated group (ATT). For the details about ATT estimation, see Appendix C.

Let $e_0(X) = P_0(D = 1 \mid X)$ denote the probability of assigning treatment 1 given covariates $X$, which is known as the *propensity score*. Throughout this study, we impose the following conditions, commonly referred to as the unconfoundedness and common support assumptions.

**Assumption 1.1.** *It holds that $(Y(1), Y(0)) \perp\!\!\!\perp D \mid X$. There exists a constant $C > 0$ independent of $n$ such that $C < e_0(x) < 1 - C$ for all $x \in \mathcal{X}$.*

When $e_0(x)$ is not constant, a distributional shift arises between the observed outcomes in the treatment and control groups, denoted by $\mathcal{G}_1$ and $\mathcal{G}_0$, respectively, where $\mathcal{G}_d := \{i \in \{1, 2, \ldots, n\} : D_i = d\}$. This shift induces bias in the

---

[1] In some cases, only treatment 1 is referred to as the treatment, while treatment 0 is referred to as the control. For simplicity, we refer to them as treatment 1 and treatment 0 throughout this study.

sample mean, $\frac{1}{|\mathcal{G}_d|} \sum_{i \in \mathcal{G}_d} Y_i = \frac{1}{|\mathcal{G}_d|} \sum_{i \in \mathcal{G}_d} Y_i(d)$, which deviates from $\mathbb{E}\left[Y(d)\right]$ and thus prevents the sample mean difference, $\frac{1}{|\mathcal{G}_1|} \sum_{i \in \mathcal{G}_1} Y_i - \frac{1}{|\mathcal{G}_0|} \sum_{i \in \mathcal{G}_0} Y_i$, from being an unbiased estimator of the ATE.

To address this issue, several debiased estimators have been proposed under standard regularity conditions. In this section, we introduce two representative estimators, the inverse probability weighting (IPW) estimator and the augmented IPW (AIPW) estimator, as follows:

**IPW estimator.** $\widetilde{\tau}^{\mathrm{IPW}} := \frac{1}{n} \sum_{i=1}^{n} \left( \frac{\mathbb{1}[D_i=1]Y_i}{e_0(X_i)} - \frac{\mathbb{1}[D_i=0]Y_i}{1-e_0(X_i)} \right) = \frac{1}{n} \sum_{i=1}^{n} \left( \frac{\mathbb{1}[D_i=1]}{e_0(X_i)} - \frac{\mathbb{1}[D_i=0]}{1-e_0(X_i)} \right) Y_i.$

**AIPW estimator.** $\widetilde{\tau}^{\mathrm{AIPW}} := \frac{1}{n} \sum_{i=1}^{n} \left( \left( \frac{\mathbb{1}[D_i=1]}{e_0(X_i)} - \frac{\mathbb{1}[D_i=0]}{1-e_0(X_i)} \right) (Y_i - \mu_0(D_i, X_i)) + \mu_0(1, X_i) - \mu_0(0, X_i) \right)$, where $\mu_0(d, X)$ is the expected conditional outcome $\mathbb{E}\left[Y(d) \mid X\right]$ of treatment $d$ given $X$. The AIPW estimator is also known as the doubly robust (DR) estimator (Bang & Robins, 2005).

**Bias-correction term.** In both estimators, the term

$$h_0(D, X) := h(D, X) := \frac{\mathbb{1}[D=1]}{e_0(X)} - \frac{\mathbb{1}[D=0]}{1-e_0(X)}$$

is crucial. This term, referred to as the *bias-correction term*, is central to ATE estimation (Schuler & van der Laan, 2024). A common approach is to estimate $e_0$ using logistic regression and then plug the resulting estimate $\widehat{e}_n^{\mathrm{L}}$ into $h$. Note that the bias-correction term is also referred to as the Riesz representer (Chernozhukov et al., 2021) or the clever covariates (van der Laan, 2006). We use the term bias-correction term because the Riesz representer is closely connected to the automatic debiased machine learning literature, and the clever covariates is closely connected to the targeted maximum likelihood estimation (TMLE) literature.

For example, in a typical one-step bias correction, we first construct an ATE estimator as $\widehat{\tau}_n^{\mathrm{DM}} := \frac{1}{n} \sum_{i=1}^{n} (\widehat{\mu}_n(1, X) - \widehat{\mu}_n(0, X))$, where $\widehat{\mu}_n$ is an estimator of $\mu_0$. This estimator is known as the direct method (DM) or naive plug-in estimator. To obtain an efficient estimator, we add the bias-correction term $\frac{1}{n} \sum_{i=1}^{n} h_0(D_i, X_i)(Y_i - \widehat{\mu}_n(D_i, X_i))$ to the first-stage DM estimator $\widehat{\tau}_n^{\mathrm{DM}}$, yielding the AIPW estimator.

In this study, we propose a method to estimate the bias-correction term, also called the Riesz representer or the clever covariates. For example, we can estimate the bias-correction term by estimating the propensity score $e_0$ using the maximum likelihood estimation. However, our interest is not in propensity score estimation but in bias-correction term estimation. As the well-known Vapnik principle states, we should avoid such an intermediate problem and ideally aim to estimate the target objective in a more direct manner (Vapnik, 1998). Following this principle, this study considers estimating $h_0(D, X)$ by directly minimizing the estimation error for the true $h_0(D, X)$.

The technical challenge is that the target objective $h_0$ is unknown. To address this issue, we employ techniques developed in the direct density-ratio estimation (DRE) literature (Sugiyama et al., 2012). In direct DRE, the goal is to minimize the empirical risk between the true density ratio and its model, even though the true density ratio is unknown. It is known that empirical risk minimization is feasible even without knowledge of the true propensity score. Since the inverse propensity score can be viewed as a density ratio, we can extend these existing methods to our setting. For causal inference researchers who are unfamiliar with DRE, we review the DRE literature in Appendix A.

Our motivation is also closely aligned with studies on Riesz regression (Chernozhukov et al., 2021) and covariate balancing weights (Imai & Ratkovic, 2013b; Deville & Särndal, 1992), which also aim to estimate the bias-correction term in a direct manner. Studies in covariate balancing focus on the balancing property of propensity score estimator and estimate them using the property. Chernozhukov et al. (2021) proposes Riesz regression which represents the bias-correction term as the Riesz representer. Although the derivation process is different, we derive the objective function that is the same as Chernozhukov et al. (2021) by using the DRE techniques. Further, we generalize our objective by using the Bregman divergence as well as DRE in Sugiyama et al. (2011). From this generalization, we further connect our approach to the covariate balancing by showing the equivalence between our objective and empirical balancing through the duality arguments discussed in Zhao (2019) and Bruns-Smith et al. (2025).

## 1.2 OUR CONTRIBUTIONS

This study has the following four contributions: (i) a general framework for directly estimating the bias-correction term (also called the Riesz representer or clever covariates) via Bregman divergence minimization; (ii) our proposed framework includes Riesz regression in Chernozhukov et al. (2021) and the tailored loss in Zhao (2019) as special cases; (iii) under our framework, we show that there are appropriate choices of bias-correction term models and

Bregman divergences under which covariate balancing is automatically realized as the dual of the Bregman divergence minimization problem (*automatic covariate balancing*); (iv) we provide a theoretical analysis of the estimator.

Our first contribution is the proposal of a framework for direct bias-correction term estimation via Bregman divergence minimization. We estimate the bias-correction term by directly minimizing the estimation error of the true bias-correction function $h_0$, measured by the Bregman divergence, $\mathrm{BR}_g^\dagger(h_0 \mid h) \coloneqq \mathbb{E}\Big[g\big(h_0(D, X)\big) - g\big(h(D, X)\big) - \partial g\big(h(d, X)\big)\big(h_0(D, X) - h(D, X)\big)\Big]$, where $g$ is a differentiable and strictly convex function. By changing $g$, we can measure the error using various metrics, such as the squared loss or KL divergence loss. Since the Bregman divergence involves the unknown function $h_0$, direct optimization is infeasible. To address this issue, we propose minimizing an alternative objective function, defined as $\mathrm{BR}_g(h) \coloneqq \mathbb{E}\Big[-g\big(h(D, X)\big) + \partial g\big(h(D, X)\big)h(D, X) - \partial g\big(h(1, X)\big) - \partial g\big(h(0, X)\big)\Big]$. Minimizing the original Bregman divergence $\mathrm{BR}_g^\dagger(h)$ is equivalent to minimizing $\mathrm{BR}_g(h)$, which does not depend on the unknown function. That is, we establish the equivalence: $h^* \coloneqq \arg\min_{h \in \mathcal{H}} \mathrm{BR}_g^\dagger(h_0 \mid h) = \arg\min_{h \in \mathcal{H}} \mathrm{BR}_g(h)$. The resulting objective function can then be approximated using an empirical risk function.

Our second contribution is the unification of existing literature. Our proposed Bregman divergence minimization objective includes Riesz regression in Chernozhukov et al. (2021) (when using the squared loss) and the tailored loss in Zhao (2019) (when using the KL divergence loss). Furthermore, our framework also integrates covariate balancing methods (Imai & Ratkovic, 2013a; Hainmueller, 2012; Zubizarreta, 2015; Chan et al., 2015; Wong & Chan, 2017). If we use linear models to approximate the bias-correction term and train the model with the squared loss (Riesz regression), the dual problem coincides with the optimization problem in stable balancing weights. If we model the bias-correction term via the propensity score with logistic models and train the model with the KL divergence loss (tailored loss), the dual problem becomes the same as the optimization problem in entropy balancing weights. Kato (2025a), a subsequent work of this study, refers to this property as *automatic covariate balancing*. See Table 1 in Section 2 and Figure 1 in Appendix.

Our third main contribution is the theoretical analysis of the estimator obtained via direct bias-correction term estimation. Since we estimate $r_0$ using empirical risk minimization, we establish bounds on the estimation error using empirical process theory. Furthermore, we present examples of ATE estimators that incorporate the bias-correction term estimated using our framework and conduct simulation studies. Using standard ATE estimation techniques, we demonstrate that our method yields a $\sqrt{n}$-consistent ATE estimator.

As a side product of our contributions, we find that we can import various existing results from the DRE literature. Since Riesz regression is essentially the same as LSIF, various results about convergence rate analysis and optimization methods have already been established. For example, Kanamori et al. (2012) shows the convergence rate when using a reproducing kernel hilbert space (RKHS) for the density ratio, or equivalently the bias-correction term. Kato & Teshima (2021) shows the rate when using neural networks, which has been further refined in Zheng et al. (2022). Rhodes et al. (2020) and Kato & Teshima (2021) point out the overfitting problem characteristic of DRE estimation and propose techniques to avoid the problem. Lin et al. (2023) finds that nearest neighbor matching can be interpreted as density ratio estimation, and it can also be interpreted as a special case of LSIF or Riesz regression (See Appendix H). These findings not only help deepen our understanding of Riesz regression, but also prevent unnecessary reinvention. For example, the covariate adaption method proposed in Chernozhukov et al. (2025) uses Riesz regression, but it is essentially the same as covariate adaption with a density ratio estimated via LSIF (Kanamori et al., 2009), except for the regression adjustment. While Chernozhukov et al. (2022a) proposes neural networks and random forests for Riesz regression, the techniques for estimating the density ratio have also been proposed in the DRE literature (Kanamori et al., 2012; Abe & Sugiyama, 2019; Rhodes et al., 2020; Kato & Teshima, 2021).

## 2 BIAS-CORRECTION TERM ESTIMATION VIA BREGMAN DIVERGENCE MINIMIZATION

In this study, we consider estimating $h_0$ by minimizing the empirical risk associated with the Bregman divergence between $h_0$ and its estimator $h\colon \{1, 0\} \times \mathcal{X} \to \mathbb{R}$.

### 2.1 POPULATION BREGMAN DIVERGENCE MINIMIZATION

Let $g\colon \mathbb{R} \to \mathbb{R}$ be a differentiable and strictly convex function. Given $d \in \{1, 0\}$, we define the Bregman divergence between $h_0$ and $h$ as $\mathrm{br}_g^\dagger\big(h_0(d, x) \mid h(d, x)\big) \coloneqq g\big(h_0(d, x)\big) - g\big(h(d, x)\big) - \partial g\big(h(d, x)\big)\big(h_0(d, x) - h(d, x)\big)$, where $\partial g$ denotes the derivative of $g$. Then, we define the average Bregman divergence as $\mathrm{BR}_g^\dagger(h_0 \mid h) \coloneqq \mathbb{E}\Big[g\big(h_0(D, X)\big) -$

$g\big(h(D,X)\big) - \partial g\big(h(d,X)\big)\big(h_0(D,X) - h(D,X)\big)\Big]$. Then, we estimate $h_0$ by $h^* = \arg\min_{h\in\mathcal{H}} \mathrm{BR}_g^\dagger\big(h_0 \mid h\big)$. By dropping the term that is irrelevant to learning, we have

$$h^* = \arg\min_{h\in\mathcal{H}} \mathrm{BR}_g\big(h\big),$$

where $\quad \mathrm{BR}_g\big(h\big) := \mathbb{E}\Big[ - g\big(h(D,X)\big) + \partial g\big(h(D,X)\big)h(D,X) - \partial g\big(h(1,X)\big) + \partial g\big(h(0,X)\big)\Big].$

This can be shown as follows:

$$h^* = \arg\min_{h\in\mathcal{H}} \sum_{d\in\{1,0\}} \mathbb{E}\Big[ \mathbb{1}[D=d]\Big(g(h_0(d,X)) - g(h(d,X)) - \partial g(h(d,X))\Big(h_0(d,X) - h(d,X)\Big)\Big)\Big]$$

$$= \arg\min_{r\in\mathcal{H}} \sum_{d\in\{1,0\}} \mathbb{E}\Big[ \mathbb{1}[D=d]\Big(-g(h(d,X)) - \partial g(h(d,x))\Big(h_0(d,X) - h(d,X)\Big)\Big)\Big]$$

$$= \arg\min_{r\in\mathcal{H}} \sum_{d\in\{1,0\}} \Big(\mathbb{E}\Big[ \mathbb{1}[D=d](-g(h(d,X)) + \partial g(h(d,X))h(d,X))\Big] - \mathbb{E}\Big[ \mathbb{1}[D=d]\partial g(h(d,x))h_0(d,X)\Big]\Big)$$

$$= \arg\min_{r\in\mathcal{H}} \Big\{\mathbb{E}\Big[ (-g(h(D,X)) + \partial g(h(D,X))h(d,X))\Big] - \mathbb{E}\Big[\partial g(h(1,X))\Big] + \mathbb{E}\Big[\partial g(h(0,X))\Big]\Big\}.$$

Here, we dropped terms irrelevant to the optimization and used $\mathbb{E}[\mathbb{1}[D=1]h_0(1,X) \mid X] = \mathbb{E}[e_0(X)h_0(1,X) \mid X] = 1$ and $\mathbb{E}[\mathbb{1}[D=0]h_0(0,X) \mid X] = -1$.

Thus, surprisingly, we demonstrate that the least squares estimate for the unknown true bias-correction term $h_0$ can be defined by an objective function that does not explicitly include $h_0$ itself. As discussed in the following subsection, this objective function can be easily approximated using observations.

## 2.2 EMPIRICAL BREGMAN DIVERGENCE MINIMIZATION

Then, we estimate the bias-correction term $h_0$ by minimizing an empirical Bregman divergence as

$$\widehat{h}_n := \arg\min_{h\in\mathcal{H}} \widehat{\mathrm{BR}}_g\big(h\big) + \lambda J(h),$$

where $J(h)$ is some regularization function and

$$\widehat{\mathrm{BR}}_g(h) := \frac{1}{n}\sum_{i=1}^{n} \Big( - g(h(D_i,X_i)) + \partial g(h(D_i,X_i))h(D_i,X_i) - \partial g(h(1,X_i) + \partial g(h(0,X_i))\Big).$$

## 2.3 LOSSES FOR THE BIAS-CORRECTION TERM ESTIMATION

By changing $g$, we can obtain various loss functions for estimating the bias-correction term, as shown in the subsequent subsections. In particular, if we use the squared loss in the Bregman divergence, we obtain Riesz regression in Chernozhukov et al. (2021), which is originally called Least-Squares Importance Fitting (LSIF) in the DRE literature Kanamori et al. (2009). Note that kernel mean matching by Gretton et al. (2009) is also the same as, or a variant of, LSIF. If we use the KL divergence, we obtain the tailored loss in Zhao (2019), which is originally called KLIEP in the DRE literature Sugiyama et al. (2008). Furthermore, as we discuss in Section 3, if we use linear models for $h_0$ and train them with the squared loss, the covariate balancing property is automatically obtained, as shown in Bruns-Smith et al. (2025). If we model $h_0$ using the propensity score $e_0$ approximated via logistic models and train it with the tailored loss, the covariate balancing property is automatically obtained, as shown in Zhao (2019). We demonstrate the correspondence of the existing methods in Table 1. Also see Figure 1 in Appendix for the relationship among bias-correction term estimation via Bregman divergence minimization, density ratio estimation, and covariate balancing, summarized in Kato (2025a) and Kato (2025c).

## 2.4 SQUARED LOSS

Our least squares method for direct bias-correction term estimation can be obtained by using a squared loss $g^{\mathrm{SL}}(h) = (h-1)^2$. By substituting this function into the Bregman divergence, we formulate the estimation problem as $h^* := \arg\min_{h\in\mathcal{H}} \mathrm{BR}_{g^{\mathrm{SL}}}(h)$, where

$$\mathrm{BR}_{g^{\mathrm{SL}}}\big(h\big) = \mathbb{E}\Big[ - 2\big(h(1,X) - h(0,X)\big) + h(D,X)^2\Big].$$

Table 1: Correspondence among DRE methods and bias-correction term estimation methods (BCE).

| DRE method | BCE method | $g(t)$ |
|---|---|---|
| LSIF (Kanamori et al., 2009) | Riesz regression (Chernozhukov et al., 2021) | $(t-1)^2/2$ |
| Kernel Mean Matching (Gretton et al., 2009) | Stable balancing weights (Zubizarreta, 2015) | |
| UKL (Nguyen et al., 2010) | Tailored loss (Zhao, 2019) | $t \log(t) - t$ |
| KLIEP (Sugiyama et al., 2008) | Entropy balancing weights (Hainmueller, 2012) | |
| Binary KL divergence | | $t \log(t) - (1+t) \log(1+t)$ |
| PULogLoss (Kato et al., 2019) | | $C \log(1-t)$ $+ Ct (\log(t) - \log(1-t))$ for $0 < t < 1$ |

Then, we estimate the bias-correction term as $\widehat{h}_n := \arg\min_{h \in \mathcal{H}} \widehat{\mathrm{BR}}_{g^{\mathrm{SL}}}(h) + \lambda J(h)$, where $\widehat{\mathrm{BR}}_{g^{\mathrm{SL}}}(h) = \frac{1}{n} \sum_{i=1}^{n} \big(-2\big(h(1, X_i) - h(0, X_i)\big) + h(D_i, X_i)^2\big)$. This objective function is the same as the one used in Chernozhukov et al. (2021). This type of estimation method is referred to as LSIF in density-ratio estimation (Kanamori et al., 2009).

### 2.5 KL DIVERGENCE LOSS

Consider $g^{\mathrm{KL}}(h) = |h| \log |h| - |h|$, which is a convex function. By substituting this function into the Bregman divergence, we formulate the estimation problem as $h^* := \arg\min_{h \in \mathcal{H}} \mathrm{BR}_{g^{\mathrm{KL}}}(h)$, where

$$\mathrm{BR}_{g^{\mathrm{KL}}}(h) := \mathbb{E}\Big[|h(D_i, X_i)| - \log(|h(1, X)|) - \log(|h(0, X)|)\Big].$$

Then, we estimate the bias-correction term as $\widehat{h}_n := \arg\min_{h \in \mathcal{H}} \widehat{\mathrm{BR}}_{g^{\mathrm{KL}}}(h) + \lambda J(h)$, where $\widehat{\mathrm{BR}}_{g^{\mathrm{KL}}}(h) = \frac{1}{n} \sum_{i=1}^{n} \big(|h(D_i, X_i)| - \log(|h(1, X_i)|) - \log(|h(0, X_i)|)\big)$. This estimation method corresponds to unnormalized Kullback–Leibler (UKL) minimization in DRE (Nguyen et al., 2010), which generalizes the KL importance estimation procedure (KLIEP). Also see Appendix B.

### 2.6 TAILORED LOSS (A VARIANT OF THE KL DIVERGENCE LOSS)

Next, as a variant of the KL divergence loss, we propose the tailored loss. Let us redefine a model $\mathcal{H}$ as a set of functions $h(1, \cdot): \mathcal{X} \to (1, \infty)$ and $h(0, \cdot): \mathcal{X} \to (-1, -\infty)$; that is, we restrict the space of $h$. This restriction is justified from the form of $h_0$ and the common support assumption. Let us consider $g^{\mathrm{TL}}(h) = (|h| - 1) \log(|h| - 1) - |h|$. By substituting this function, we obtain

$$\mathrm{BR}_{g^{\mathrm{TL}}}(h) := \mathbb{E}\Big[\log(|h(D, X)| - 1) + |h(D, X)| - \log(|h(1, X)| - 1) - \log(|h(0, X)| - 1)\Big].$$

Note that it holds that $\mathrm{BR}_{g^{\mathrm{TL}}}(h) := \mathbb{E}\Big[-\mathbb{1}[D = 0] \log(|h(1, X)| - 1) - \mathbb{1}[D = 1] \log(|h(0, X)| - 1) + \mathbb{1}[D = 1]h(1, X) - \mathbb{1}[D = 0]h(0, X)\Big]$. Then, we estimate the bias-correction term as $\widehat{h}_n := \arg\min_{h \in \mathcal{H}} \widehat{\mathrm{BR}}_{g^{\mathrm{TL}}}(h)$, where the empirical Bregman divergence becomes $\widehat{\mathrm{BR}}_{g^{\mathrm{TL}}}(h) = \frac{1}{n} \sum_{i=1}^{n} \big(\mathbb{1}[D_i = 0] \log(|h(1, X_i)| - 1) + \mathbb{1}[D_i = 1] \log(|h(0, X_i)| - 1) + \mathbb{1}[D_i = 1]|h(1, X_i)| - \mathbb{1}[D_i = 0]|h(0, X_i)|\big)$.

## 3 AUTOMATIC COVARIATE BALANCING

Under specific choices of Riesz regression models and Bregman divergence, we can automatically enforce the covariate balancing property. The key tool is the duality relationship between the Bregman divergence minimization problem and the covariate balancing optimization problem. This result is shown in Kato (2025a), and we introduce the result for reference.

### 3.1 LINEAR MODELS AND SQUARED LOSS

Consider a linear model

$$h_\beta(D, X) = \Phi(D, X)^\top \beta,$$

where $\Phi: \{1, 0\} \times \mathcal{X} \to \mathbb{R}^p$ is a basis function. For this model, using the squared loss (Riesz regression) automatically achieves covariate balancing, as discussed in Bruns-Smith et al. (2025).

Specifically, under linear models, by duality, this MSE minimization problem is equivalent to solving

$$\min_{w \in \mathbb{R}^n} \|w\|_2^2 \quad \text{s.t.} \quad \sum_{i=1}^n w_i \Phi(D_i, X_i) - \left( \sum_{i=1}^n \left( \Phi(1, X_i) - \Phi(0, X_i) \right) \right) = \mathbf{0}_p,$$

where $\mathbf{0}_p$ is the $p$-dimensional zero vector. This optimization problem matches that used to obtain stable weights (Zubizarreta, 2015).

It enforces the covariate balancing condition $\sum_{i=1}^n \widehat{w}_i \Phi(D_i, X_i) - \left( \sum_{i=1}^n \left( \Phi(1, X_i) - \Phi(0, X_i) \right) \right) = \mathbf{0}_p$, where $\widehat{w}_i = \Phi(D_i, X_i)^\top \widehat{\beta}$.

Another advantage of using linear models is that we can write the entire ATE estimation with a single linear model, as shown by Bruns-Smith et al. (2025).

### 3.2 LOGISTIC MODELS AND TAILORED LOSS

We can model the Riesz representer by modeling the propensity score as

$$h_\beta(D, X) = \mathbb{1}[D = 1] r_\beta(1, X) - \mathbb{1}[D = 0] r_\beta(0, X),$$

where $r_\beta(1, X) = \frac{1}{e_\beta(X)}$, $r_\beta(0, X) = \frac{1}{1 - e_\beta(X)}$, $e_\beta(X) := \frac{1}{1 + \exp\left( -\beta^\top \Phi(X) \right)}$, and $\Phi \colon \mathcal{X} \to \mathbb{R}^p$ is a basis function.

Note that we do not include $D$, unlike the basis function used in linear models. For this model, if we use the KL-divergence–flavored convex function defined in Section 2.6, which corresponds to the tailored loss in Zhao (2019), we automatically achieve covariate balancing.

Define $\widehat{\beta} := \arg\min_\beta \frac{1}{n} \sum_{i=1}^n \sum_{d \in \{1,0\}} \left( \mathbb{1}[D_i = d] \left( -\log \left( \frac{1}{r_\beta(d, X_i) - 1} \right) + r_\beta(d, X_i) \right) \right)$, and denote $r_{\widehat{\beta}}$ by $\widehat{r}$. Under logistic models, by duality, the KL divergence-flavored loss is equivalent to solving

$$\min_{w \in (1, \infty)^n} \sum_{i=1}^n (w_i - 1) \log(w_i - 1) \quad \text{s.t.} \quad \left( \sum_{i=1}^n \left( \mathbb{1}[D_i = 1] w_i \Phi(X_i) - \mathbb{1}[D_i = 0] w_i \Phi(X_i) \right) \right) = \mathbf{0}_p.$$

This optimization problem matches that used in entropy balancing (Hainmueller, 2012). Note that this objective function is derived from $\widehat{\mathrm{BR}}_{g^{\mathrm{TL}}}(h)$ when we use the logistic model specified in this section.

As a result, we obtain $\sum_{i=1}^n \left( \mathbb{1}[D_i = 1] \widehat{w}_i \Phi(X_i) - \mathbb{1}[D_i = 0] \widehat{w}_i \Phi(X_i) \right) = \mathbf{0}_p$, where $\widehat{w}_i = \widehat{r}(X_i)$.

This model has the advantage that we can use a basis function $\Phi(X)$ independent of $D$. Moreover, it naturally achieves covariate balance in the sense that the covariate distributions match between the treated and control groups. Additionally, it allows us to automatically impose nonnegativity on $h(1, X)$ and $-h(0, X)$, which may be violated in linear models. Note that $h_0(1, X) = \frac{1}{e(X)}$ and $h_0(1, X) = \frac{1}{1 - e(X)}$.

### 3.3 COMPARISON

We first discuss the advantages of using logistic models over linear models. One benefit of using logistic models is that we can simplify the basis function by making it independent of $D$. Furthermore, we can express covariate balancing in a clearer form as $\sum_{i=1}^n \left( \mathbb{1}[D_i = 1] \widehat{w}_i \Phi(X_i) - \mathbb{1}[D_i = 0] \widehat{w}_i \Phi(X_i) \right) = \mathbf{0}_p$, while under linear models, $\sum_{i=1}^n \widehat{w}_i \Phi(D_i, X_i) - \left( \sum_{i=1}^n \left( \Phi(1, X_i) - \Phi(0, X_i) \right) \right) = \mathbf{0}_p$ is attained, but it is somewhat harder to interpret. Moreover, using logistic models incorporates more information about the form of the bias-correction term, which includes the inverse propensity function. Logistic models also naturally impose restrictions such that $h(1, X) \in (1, \infty)$ and $h(0, X) \in (-\infty, -1)$ under the common support assumption.

In contrast, if we use linear models, we can express the entire ATE estimator with a single linear model, as shown in Bruns-Smith et al. (2025). Furthermore, we can obtain the estimator of the bias-correction term as a closed-form solution. In addition, as discussed in Kato (2025b), a subsequent work of this study. nearest neighbor matching is also an instance of linear models trained via Riesz regression (squared loss). We introduce the result in Appendix H for reference.

Ultimately, there is no clear dominance between the use of linear and logistic models. Moreover, we can also use more complex models, such as random forests and neural networks. The choice of model should be made based on the

data and application, and once the model is selected, we can determine appropriate specifications that ensure covariate balancing automatically.

## 4 ESTIMATION ERROR ANALYSIS

This section provides an estimation error analysis for $h_0$ estimated by the direct bias-correction term estimation method. We can use various models for $\mathcal{H}$, including RKHS and neural networks.

### 4.1 MODEL

We define a model of the bias-correction term $h_0$ by $h(D, X) = \zeta^{-1} \circ f(D, X) = \zeta^{-1}(f(D, X))$, where $\zeta$ is a continuously differentiable and globally Lipschitz link function, and $f$ is some basic model. For example, if we use linear model for the bias-correction term $h_0$, we can write $h(D, X) = \Phi(D, X)^{\top}\beta$, where $\zeta$ is the identity function, $f(D, X) = \Phi(D, X)^{\top}\beta$, $\Phi$ is some basis function and $\beta$ is the corresponding parameter. If we use logistic model for the bias-correction term $h_0$, we can use logistic link for $\zeta$, and $f(D, X) = \Phi(X)^{\top}\beta$.

### 4.2 RKHS

First, we investigate the case with RKHS regression. Let $\mathcal{F}^{\mathrm{RKHS}}$ be a class of RKHS functions, and define $\widehat{f}_n^{\mathrm{RKHS}} := \arg\min_{f \in \mathcal{F}^{\mathrm{RKHS}}} \widehat{\mathcal{L}}_n(\zeta^{-1} \circ f) + \lambda\|f\|_{\mathcal{F}}^2$, where $\|\cdot\|_{\mathcal{F}}^2$ is the RKHS norm. Then, we define an estimator as $h^{\mathrm{RKHS}} := \zeta^{-1} \circ \widehat{f}_n^{\mathrm{RKHS}}$ We analyze the estimation error by employing the results in Kanamori et al. (2012), which study RKHS-based LSIF in DRE. We define the following localized class of RKHS functions as a technical device: $\mathcal{F}_M^{\mathrm{RKHS}} := \{f \in \mathcal{F}^{\mathrm{RKHS}}: I(f) \le M\}$ for some norm $I(f)$ of $f$. We also define $\mathcal{H}^{\mathrm{RKHS}} := \{\zeta^{-1} \circ f: f \in \mathcal{F}^{\mathrm{RKHS}}\}$. We then make the following assumption using this localized class.

**Assumption 4.1.** *There exist constants $0 < \gamma < 2$, $0 \le \beta \le 1$, $c_0 > 0$, and $A > 0$ such that for all $M \ge 1$, it holds that $H_B(\delta, \mathcal{F}_M^{RKHS}, P_0) \le A\left(\frac{M}{\delta}\right)^{\gamma}$, where $H_B(\delta, \mathcal{F}_M^{RKHS}, P_0)$ is the bracketing entropy with radius $\delta > 0$ for the function class $\mathcal{F}_M^{RKHS}$ and the distribution $P_0$.*

For the details of the definition of the bracketing entropy, see Appendix F and Definition 2.2 in van de Geer (2000).

Under these preparations, we establish an estimation error bound.

**Theorem 4.1** ($L_2$-norm estimation error bound). *Suppose that $g$ is $\mu$-strongly convex and there exist constant $C > 0$ such that $|g''(t)| \le C \quad \forall t \in \mathbb{R}$. Assume also that $\zeta^{-1}(0)$ is finite. Suppose that Assumptions 1.1 and 4.1 hold. Set the regularization parameter $\lambda = \lambda_n$ so that $\lim_{n\to\infty} \lambda_n = 0$ and $\lambda_n^{-1} = O(n^{1-\delta})$ $(n \to \infty)$. If $h_0 \in \mathcal{H}^{RKHS}$, then we have $\left\|\widehat{h}_n^{RKHS}(D, X) - h_0(D, X)\right\|_{L_2(P_0)}^2 = O_{P_0}\left(\lambda^{1/2}\right).$*

The proof is provided in Appendix F, following the approach of Kanamori et al. (2012). The parameter $\gamma$ is determined by the function class to which $f_0$ belongs.

### 4.3 NEURAL NETWORKS

Second, we provide an estimation error analysis when we use neural networks for $\mathcal{H}$. Our analysis is mostly based on Kato & Teshima (2021) and Zheng et al. (2022). We define Feedforward neural networks (FNNs) as follows:

**Definition 4.1** (FNNs. From Zheng et al. (2022)). *Let $\mathcal{D}$, $\mathcal{W}$, $\mathcal{U}$, and $\mathcal{S} \in (0, \infty)$ be parameters that can depend on $n$. Let $\mathcal{F}^{\mathrm{FNN}} := \mathcal{F}_{M,\mathcal{D},\mathcal{W},\mathcal{U},\mathcal{S}}^{\mathrm{FNN}}$ be a class of ReLU-activated FNNs with parameter $\theta$, depth $\mathcal{D}$, width $\mathcal{W}$, size $\mathcal{S}$, number of neurons $\mathcal{U}$, satisfies the following conditions: (i) the number of hidden layers is $\mathcal{D}$; (ii) the maximum width of the hidden layers is $\mathcal{W}$; (iii) the number of neurons in $e_\theta$ is $\mathcal{U}$; (iv) the total number of parameters in $e_\theta$ is $\mathcal{S}$.*

For the model $\mathcal{F}^{\mathrm{FNN}}$, we define $\widehat{f}_n^{\mathrm{FNN}} := \arg\min_{f \in \mathcal{F}^{\mathrm{FNN}}} \widehat{\mathcal{L}}_n(\zeta^{-1} \circ f)$. Then, we define an estimator as $\widehat{h}_n^{\mathrm{FNN}} := \zeta^{-1} \circ \widehat{f}_n^{\mathrm{FNN}}$.

For the estimator, we can prove an estimation error bound. Let us make the following assumption.

**Assumption 4.2.** *There exists a constant $0 < M < \infty$ such that $\|f_0\|_\infty < M$, and $\|f\|_\infty \le M$ for any $f \in \mathcal{F}^{FNN}$.*

Let $\mathrm{Pdim}(\mathcal{F}^{\mathrm{FNN}})$ be the pseudo-dimension of $\mathcal{F}^{\mathrm{FNN}}$. For the definition, see Anthony & Bartlett (1999) and Definition 3 in Zheng et al. (2022). Then, we prove the following estimation error bound:

**Theorem 4.2** (Estimation error bound for neural networks). *Suppose that $g$ is $\mu$-strongly convex and there exist constant $C > 0$ such that $|g''(t)| \leq M \quad \forall t \in \mathbb{R}$. Assume also that $\zeta^{-1}(0)$ is finite. Suppose that Assumption 4.2 holds. For $f_0$ such that $h_0 = \zeta^{-1} \circ f_0$, also assume $f_0 \in \Sigma(\beta, M, [0,1]^d)$ with $\beta = k + a$, where $k \in \mathbb{N}^+$ and $a \in (0,1]$, and $\mathcal{F}^{\text{FNN}}$ has width $\mathcal{W}$ and depth $\mathcal{D}$ such that $\mathcal{W} = 38(\lfloor \beta \rfloor + 1)^2 d^{\lfloor \beta \rfloor + 1}$ and $\mathcal{D} = 21(\lfloor \beta \rfloor + 1)^2 \lceil n^{\frac{d}{2(d+2\beta)}} \log_2\left(8n^{\frac{d}{2(d+2\beta)}}\right)\rceil$. Then, for $M \geq 1$ and $n \leq \text{Pdim}(\mathcal{F}^{\text{FNN}})$, it holds that*
$$\left\|\widehat{h}_n^{\text{FNN}}(D,X) - h_0(D,X)\right\|_{L_2(P_0)}^2 = C_0(\lfloor \beta \rfloor + 1)^9 d^{2\lfloor \beta \rfloor + (\beta \wedge 3)} n^{-\frac{2\beta}{d+2\beta}} \log^3 n, \text{ where } C_0 > 0 \text{ is a constant independent of } n.$$

The proof is provided in Appendix G, following the approach of Zheng et al. (2022). This result directly implies the minimax optimality of the proposed method when $f_0$ belongs to a Hölder class.

## 5 EXAMPLE ABOUT THE AIPW ESTIMATOR

This section introduces the AIPW estimator with nuisance parameters estimated using our proposed direct bias-correction term estimation. We prove that under certain conditions, the proposed estimator is asymptotically normal. Note that this result is well known in the literature except for the use of nuisance parameters estimated via our direct bias-correction term estimation. The purpose of this section is not to provide novel methodological or theoretical results but to present an application of our proposed method.

We analyze the AIPW estimator with an estimated propensity score. Recall that the AIPW estimator is defined as $\widetilde{\tau}_n^{\text{AIPW}} = \frac{1}{n} \sum_{i=1}^n \left(\widehat{h}_n(D_i, X_i)(Y_i - \widehat{\mu}_n(D_i, X_i)) + \widehat{\mu}_n(1, X_i) - \widehat{\mu}_n(0, X_i)\right)$, which is also called the DR estimator.

We first make the following assumption.

**Assumption 5.1** (Donsker condition or cross fitting). *Either of the followings holds: (i) the hypothesis classes $\mathcal{H}$ and $\mathcal{M}$ belong to the Donsker class, or (ii) $\widehat{\mu}_n$ and $\widehat{h}_n$ are estimated via cross fitting.*

For example, the Donsker condition holds when the bracketing entropy of $\mathcal{H}$ is finite. In contrast, it is violated in high-dimensional regression or series regression settings where the model complexity diverges as $n \to \infty$. For neural networks, the assumption holds if both the number of layers and the width are finite. However, if these quantities grow with the sample size, the assumption is no longer valid.

Even if the Donsker condition does not hold, we can still establish asymptotic normality by employing sample splitting (Klaassen, 1987). There are various ways to implement sample splitting, and one of the most well-known is cross-fitting, used in double machine learning (DML, Chernozhukov et al., 2018). In DML, the dataset is split into several folds, and the nuisance parameters are estimated using only a subset of the folds. This ensures that in $\widehat{h}_n(D_i, X_i)(Y_i - \widehat{\mu}_n(D_i, X_i)) + \widehat{\mu}_n(1, X_i) - \widehat{\mu}_n(0, X_i)$, the observations $(X_i, D_i, Y_i)$ are not used to construct $\widehat{\mu}_n$ and $\widehat{r}_n$. For more details, see Chernozhukov et al. (2018).

**Assumption 5.2** (Convergence rate). *$\left\|\widehat{h} - h_0\right\|_2 = o_p(1)$, $\left\|\widehat{\mu} - \mu_0\right\|_2 = o_p(1)$, and $\left\|\widehat{h} - h_0\right\|_2 \left\|\widehat{\mu} - \mu_0\right\|_2 = o_p(1/\sqrt{n})$.*

Under these assumptions, we show the asymptotic normality of $\widetilde{\tau}_n^{\text{AIPW}}$. We omit the proof. For details, see Schuler & van der Laan (2024), for example.

**Theorem 5.1** (Asymptotic normality). *Suppose that Assumptions 1.1, and 5.1–5.2 hold. Then, the AIPW estimator converges in distribution to a normal distribution as $\sqrt{n}\left(\widetilde{\tau}_n^{\text{AIPW}} - \tau_0\right) \xrightarrow{\text{d}} \mathcal{N}(0, V^*)$, where $V^*$ is the efficiency bound defined as $V^* := \mathbb{E}\left[\frac{\sigma^2(1,X)}{e_0(X)} + \frac{\sigma^2(0,X)}{1-e_0(X)} + \left(\tau_0(X) - \tau_0\right)^2\right]$ and $\tau_0(X) := \mathbb{E}[Y(1) - Y(0) \mid X]$.*

Here, $V^*$ matches the efficiency bound given as the variance of the efficient influence function (van der Vaart, 1998). Thus, this estimator is efficient.

### 5.1 COMPARISON WITH THE STANDARD DRE APPROACHES

If we follow the standard DRE approach, we may formulate the problem as the direct estimation of $r_0(1, X)$. For example, when using LSIF, the risk is given by $\mathbb{E}\left[-2r(1,X)\right] + \mathbb{E}\left[\mathbb{1}[D=1]r(1,X)^2\right]$, which corresponds to a part of our risk: $\mathbb{E}\left[-2r(1,X) - 2r(0,X) + \mathbb{1}[D=1]r(1,X)^2 + \mathbb{1}[D=0]r(0,X)^2\right]$. Thus, our proposed method is closely

Table 2: Experimental results. We report the empirical MSE and Bias of each method.

| Data | Dimension | | DM | DBC (LS) | | DBC (KL) Three-layer perceptron | | MLE | | CBPS | | RieszNet Dragonnet | | | DM Linear model |
|---|---|---|---|---|---|---|---|---|---|---|---|---|---|---|---|
| | | | | IPW | DR | IPW | DR | IPW | DR | IPW | DR | IPW | DM | DR | |
| Model 1 | $K=3$ | MSE | 0.006 | 0.392 | 0.005 | 0.374 | 0.005 | 0.330 | 0.004 | 1.429 | 0.006 | 0.017 | 0.021 | 0.040 | 2.781 |
| | $K=3$ | Bias | -0.037 | -0.299 | -0.024 | -0.316 | -0.023 | -0.257 | -0.022 | -0.747 | -0.037 | -0.027 | -0.025 | -0.053 | -0.197 |
| | $K=3$ | MSE | 0.521 | 1.956 | 0.481 | 2.779 | 0.478 | 6.510 | 0.507 | 3.570 | 0.515 | 0.464 | 0.510 | 0.379 | 7.511 |
| | $K=10$ | Bias | 0.094 | -0.930 | 0.086 | -0.822 | 0.088 | -0.268 | 0.091 | -1.422 | 0.089 | -0.093 | -0.106 | -0.017 | 0.101 |
| Model 2 | $K=3$ | MSE | 0.048 | 0.343 | 0.033 | 0.819 | 0.037 | 2.838 | 0.045 | 1.848 | 0.044 | 0.030 | 0.034 | 0.051 | 2.866 |
| | $K=3$ | Bias | -0.009 | -0.275 | -0.011 | -0.382 | -0.010 | -0.403 | -0.011 | -0.781 | -0.012 | -0.022 | -0.020 | -0.057 | -0.214 |
| | $K=3$ | MSE | 0.517 | 2.006 | 0.474 | 2.980 | 0.477 | 6.517 | 0.507 | 3.816 | 0.512 | 0.407 | 0.446 | 0.424 | 7.482 |
| | $K=10$ | Bias | 0.085 | -0.944 | 0.082 | -0.823 | 0.085 | -0.269 | 0.089 | -1.410 | 0.084 | -0.087 | -0.096 | -0.012 | 0.093 |

connected to LSIF. However, the standard DRE approach does not address whether it is suitable for bias-correction term estimation. In fact, we can estimate $r_0$ by minimizing the LSIF risk, but our proposed method adopts a different risk: the sum of $\mathbb{E}\big[-2r(1, X)\big] + \mathbb{E}\big[\mathbb{1}[D=1]r(1, X)^2\big]$ and $\mathbb{E}\big[-2r(0, X)\big] + \mathbb{E}\big[\mathbb{1}[D=0]r(0, X)^2\big]$, which is directly related to the bias-correction term.

## 6 SIMULATION STUDIES

We assess the performance of our method through simulation studies, evaluating ATE estimation error. We denote our direct bias-correction term estimation methods as DBC (LS) when using the squared loss, and DBC (TL) when using the tailored loss. We compare our approach with ATE estimators using propensity score estimated by maximum likelihood estimation (MLE), CBPS (Imai & Ratkovic, 2013a), and RieszNet (Chernozhukov et al., 2022a). Because our DBC (LS) is equivalent to Resz regression, we include RieszNet primarily as a numerical check of equivalence, noting architectural differences. In this section, for simplicity, we do not apply cross-fitting. We also conduct experiments in Appendices I and J using synthetic and semi-synthetic data, respectively, in which we apply cross-fitting.

We consider two different dimensions for $X$, setting $K=3$ and $K=10$, and two different outcome models. This results in a total of four experimental settings. In all cases, the true ATE is fixed at $\tau_0 = 5.0$. To generate synthetic data, we first sample covariates $X_i$ from a multivariate normal distribution $\mathcal{N}(0, I_K)$, where $I_K$ denotes the $K \times K$ identity matrix. The propensity score is then defined as $e_0(X_i) = \frac{1}{1+\exp\left(-h(X_i)\right)}$, where $h(X_i) = \sum_{j=1}^{3} \alpha_j X_{i,j} + \sum_{j=1}^{3} \beta_j X_{i,j}^2 + \gamma_1 X_{i,1}X_{i,2} + \gamma_2 X_{i,2}X_{i,3} + \gamma_3 X_{i,1}X_{i,3}$. The coefficients $\alpha_j$, $\beta_j$, and $\gamma_j$ are independently drawn from $\mathcal{N}(0, 0.5)$. Given these propensity scores, the treatment assignment $D$ is sampled accordingly. The outcome is then generated under two models, referred to as Model 1 and Model 2. In Model 1, we specify $Y_i = \big(X_i^\top \beta\big)^2 + 1.1 + \tau_0 D_i + \varepsilon_i$, where $\varepsilon_i \sim \mathcal{N}(0, 1)$ and $\tau_0 = 5.0$. In Model 2, the outcome is generated as $Y_i = X_i^\top \beta + \big(X_i^\top \beta\big)^2 + 3\sin(X_{i,1}) + 1.1 + \tau_0 D_i + \varepsilon_i$.

We model $h_0$ by modeling $e_0$. To model $e_0$, we use a three-layer neural network with an Exponential Linear Unit (ELU) activation function for each hidden layer (100 nodes per layer). The final output layer applies a sigmoid function to ensure that the estimated propensity scores remain in $(0, 1)$. We use this model for our method, logistic regression, and CBPS. For RieszNet, we adopt the DragonNet architecture proposed in Shi et al. (2019), following Chernozhukov et al. (2022b). For each method, including ours, we compute both the IPW and AIPW estimators using the estimated scores. Additionally, we include the direct method (DM) estimator with neural networks for comparison. In each case, the expected conditional outcomes are estimated using a three-layer neural network (100 nodes per hidden layer, with ELU activation). As a baseline, we also consider the DM estimator with linear models.

The sample size is fixed at $n = 3000$. As noted earlier, we evaluate two values of $K$ ($K = 3$ and $K = 10$) and two outcome-model specifications (Model 1 and Model 2), resulting in four experimental configurations. Each setting is repeated 500 times. We report the MSEs and biases of the resulting ATE estimates in Table 2 for $n = 3000$. Overall, the results indicate that our direct bias-correction approach achieves competitive or superior estimation accuracy compared with logistic regression and CBPS, highlighting the benefits of explicitly estimating the bias-correction term in the ATE context. RieszNet tends to outperform our method, but we consider this to be partly due to differences in the regression models. While RieszNet employs DragonNet, we use a simpler implementation. We do not employ such models, as model complexity is not our primary focus. Nevertheless, we emphasize that our method outperforms most existing approaches while exhibiting comparable performance to RieszNet.

## 7 CONCLUSION

This study proposed direct bias-correction term estimation in ATE estimation. Instead of focusing on estimating the propensity score itself, our approach directly minimizes the estimation error of the bias-correction term, leveraging empirical risk minimization techniques. We demonstrated that this direct approach enhances estimation accuracy by avoiding the intermediate step of propensity score estimation. Additionally, our method was analyzed through the lens of Bregman divergence minimization, providing a generalized framework.

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

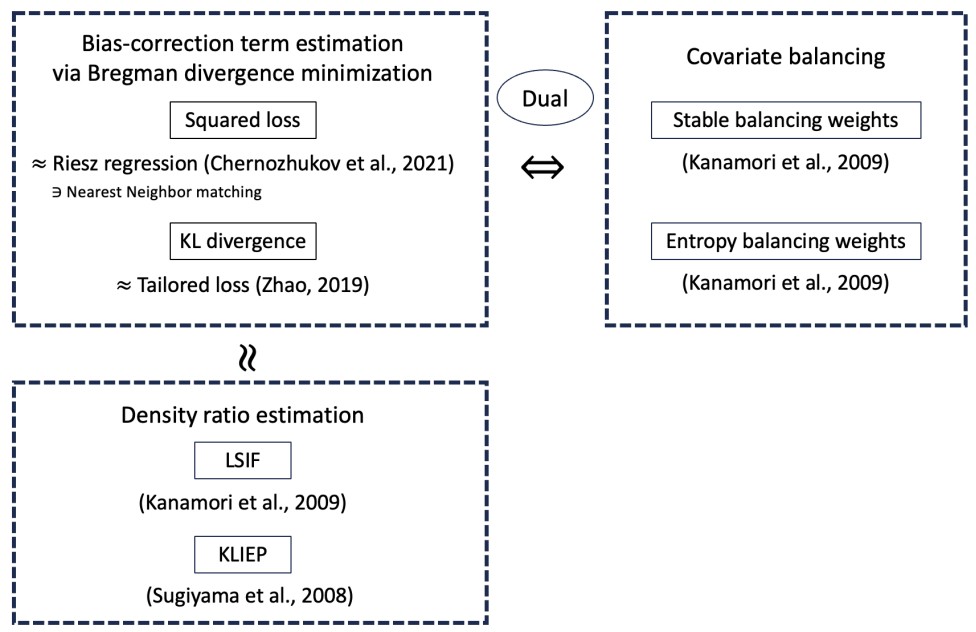

Figure 1: Relationship among bias-correction term estimation via Bregman divergence minimization, density ratio estimation, and covariate balancing. This figure is made using the results in Kato (2025a) and Kato (2025c).

## A  DENSITY-RATIO ESTIMATION (DRE)

Given two probability distributions $P$ and $Q$ over a common space $\mathcal{X}$, the density ratio function is defined as

$$r_0(x) := \frac{p(x)}{q(x)},$$

where $p(x)$ and $q(x)$ denote the density functions of $P$ and $Q$, respectively. DRE is a fundamental problem in statistical learning, with applications in importance sampling, anomaly detection, and covariate shift adaptation.

In DRE, estimating the two densities separately can magnify estimation errors, whereas directly modeling and estimating the density ratio can lead to improved accuracy. Thus, the aim of DRE is to estimate the density ratio in an end-to-end manner by directly optimizing a single objective. Various methods for DRE have been proposed (Huang et al., 2007; Gretton et al., 2009; Qin, 1998; Cheng & Chu, 2004; Nguyen et al., 2010; Kato et al., 2019), many of which can be generalized as instances of Bregman divergence minimization (Sugiyama et al., 2011; Kato & Teshima, 2021).

Let $\mathcal{R}$ be a hypothesis class for $r_0$, consisting of functions $r \colon \mathcal{X} \to \mathbb{R}$. The goal of direct DRE is to find an optimal function $r^* \in \mathcal{R}$ that best approximates $r_0$. A natural approach is to minimize the expected squared error:

$$\mathbb{E}_P \left[ \left( r_0(X) - r(X) \right)^2 \right].$$

However, since $r_0(x)$ is unknown, direct minimization of this objective is infeasible.

Instead, we derive an equivalent formulation that does not require knowledge of $r_0$. Specifically, we show that minimizing the expected squared error is equivalent to minimizing the following alternative objective:

$$-2\mathbb{E}_Q \left[ r(X) \right] + \mathbb{E}_P \left[ r(X)^2 \right].$$

This transformation enables empirical risk minimization without explicit access to the true density ratio.

Furthermore, we extend this framework by providing theoretical guarantees on the estimation error using tools from empirical process theory. From the perspective of Bregman divergence minimization, we establish a generalized methodology for DRE that accommodates various estimation strategies.

Finally, we present numerical experiments that demonstrate the effectiveness of our approach in practical scenarios, including importance weighting and outlier detection.

## B  SILVERMAN'S TRICK

Note that minimization of the Bregman divergence with the KL divergence loss is equal to

$$r^* = \arg\max_{r \in \mathcal{R}} \sum_{d \in \{1,0\}} \mathbb{E}\left[\log r(d, X)\right] \quad \text{s.t.} \quad \mathbb{E}\left[\mathbb{1}[D = 1]r(1, X_i)\right] = \mathbb{E}\left[\mathbb{1}[D = 0]r(0, X_i)\right] = 1.$$

This technique is known as Silverman's trick (Silverman, 1982). For details, see Theorem 3.3 in Kato et al. (2023). We can replace the expected values with the sample means and define the estimation problem as $\widehat{r}_n = \arg\max_{r \in \mathcal{R}} \frac{1}{n} \sum_{i=1}^n \sum_{d \in \{1,0\}} \log r(d, X_i)$ s.t. $\frac{1}{n} \sum_{i=1}^n \mathbb{1}[D_i = 1]r(1, X_i) = \frac{1}{n} \sum_{i=1}^n \mathbb{1}[D_i = 0]r(0, X_i) = 1$.

## C  ESTIMATION OF THE AVERAGE TREATMENT EFFECT FOR THE TREATED (ATT)

Our method can also be applied to other estimands, such as the ATT, which is defined as

$$\alpha_0 := \mathbb{E}\left[Y(1) - Y(0) \mid D = 1\right].$$

The IPW and AIPW estimators designed for the ATT are given by

**IPW estimator.** $\widetilde{\alpha}^{\text{IPW}} := \frac{1}{n} \sum_{i=1}^n \left( \frac{\mathbb{1}[D_i = 1]Y_i}{\pi_0} - \frac{e_0(X_i)\mathbb{1}[D_i = 0]Y_i}{\pi_0(1 - e_0(X_i))} \right) = \frac{1}{n} \sum_{i=1}^n \left( \frac{\mathbb{1}[D = 1]}{\pi_0} - \frac{e_0(X)\mathbb{1}[D = 0]}{\pi_0(1 - e_0(X))} \right) Y_i.$

**AIPW estimator.** $\widetilde{\alpha}^{\text{AIPW}} := \frac{1}{n} \sum_{i=1}^n \left( \frac{\mathbb{1}[D = 1]}{\pi_0} - \frac{e_0(X)\mathbb{1}[D = 0]}{\pi_0(1 - e_0(X))} \right) (Y_i - \mu_0(0, X_i)),$

where $\pi_0 = \mathbb{E}[\mathbb{1}[D = 1]]$.

Thus, the bias-correction term for ATT estimation is given as

$$\widetilde{h}_0(D, X) := \frac{\mathbb{1}[D = 1]}{\pi_0} - \frac{e_0(X)\mathbb{1}[D = 0]}{\pi_0(1 - e_0(X))},$$

where $\pi_0 = \mathbb{E}[\mathbb{1}[D = 1]]$.

Let $w_0(x) := \frac{e_0(X)}{(1 - e_0(X))}$. Then, we denote the bias-correction term as

$$\widetilde{h}_0(D, X) := \frac{\mathbb{1}[D = 1]}{\pi_0} - \frac{w_0(X)\mathbb{1}[D = 0]}{\pi_0}.$$

Let $\mathcal{W}$ be a set of functions $w \colon \mathcal{X} \to \mathbb{R}_+$. Then, we define the following least squares:

$$w^* := \arg\min_{r \in \mathcal{R}} \mathbb{E}\left[ \left( \widetilde{h}(D, X; r_0, \pi_0) - \widetilde{h}(D, X; r, \pi_0) \right)^2 \right].$$

Note that we use $\pi_0$ itself. We can show that this least squares is equivalent to

$$w^* = \arg\min_{r \in \mathcal{R}} \left\{ -2\mathbb{E}\left[w(X)\right] + \mathbb{E}\left[w(X)^2\mathbb{1}[D = 0]\right] \right\},$$

where $\mathbb{E}_1$ is expectation over the treated group ($p(x \mid d = 1)$). The empirical version of this risk is given as

$$\widehat{w} := \arg\min_{r \in \mathcal{R}} \left\{ -2\frac{1}{\sum_{i=1}^n \mathbb{1}[D_i = 1]} \sum_{i=1}^n \mathbb{1}[D_i = 1]w(X_i) + \frac{1}{n} \sum_{i=1}^n w(X_i)^2 \right\},$$

We can demonstrate the equivalence between the two least-squares formulations as follows:

$$
\begin{aligned}
w^* &= \arg\min_{r \in \mathcal{R}} \mathbb{E}\left[ \left( \widetilde{h}(D, X; r_0, \pi_0) - \widetilde{h}(D, X; r, \pi_0) \right)^2 \right] \\
&= \arg\min_{r \in \mathcal{R}} \mathbb{E}\left[ \left( w_0(X)\mathbb{1}[D = 0] - w(X)\mathbb{1}[D = 0] \right)^2 \right] \\
&= \arg\min_{r \in \mathcal{R}} \mathbb{E}\left[ -2w_0(X)w(X)\mathbb{1}[D = 0] + w(X)^2\mathbb{1}[D = 0] \right].
\end{aligned}
$$

To see this equivalence, consider

$$
\begin{aligned}
&\mathbb{E}\left[w_0(X)w(X)\mathbb{1}[D=0]\right] \\
&= \mathbb{E}\left[\mathbb{E}\left[w_0(X)w(X)(1-e_0(X))\right]\right] \\
&= \mathbb{E}\left[e_0(X)w(X)/\pi_0\right] \\
&= \int \frac{1}{\pi_0}e_0(x)w(x)p_0(x)\mathrm{d}x \\
&= \int \frac{1}{\pi_0}\frac{\pi_0 p_0(x \mid d=1)}{p_0(x)}w(x)p_0(x)\mathrm{d}x \\
&= \int p_0(x \mid d=1)w(x)\mathrm{d}x.
\end{aligned}
$$

This confirms the equivalence between the two least-squares objectives.

## D   PRELIMINARY

This section introduces notions that are useful for the theoretical analysis.

### D.1   RADEMACHER COMPLEXITY

Let $\sigma_1, \ldots, \sigma_n$ be $n$ independent Rademacher random variables; that is, independent random variables for which $P(\sigma_i = 1) = P(\sigma_i = -1) = 1/2$. Let us define

$$
\mathfrak{R}_n f := \frac{1}{n}\sum_{i=1}^{n}\sigma_i f(W_i).
$$

Additionally, given a class $\mathcal{F}$, we define

$$
\mathfrak{R}_n \mathcal{F} := \sup_{f \in \mathcal{F}} \mathfrak{R}_n f.
$$

Then, we define the Rademacher average as $\mathbb{E}[\mathfrak{R}_n \mathcal{F}]$ and the empirical Rademacher average as $\mathbb{E}_\sigma[\mathfrak{R}_n \mathcal{F} \mid X_1, \ldots, X_n]$.

### D.2   LOCAL RADEMACHER COMPLEXITY BOUND

Let $\mathcal{F}$ be a class of functions that map $\mathcal{X}$ into $[a, b]$. For $f \in \mathcal{F}$, let us define

$$
Pf := \mathbb{E}[f(W)],
$$

$$
P_n f := \frac{1}{n}\sum_{i=1}^{n}f(W_i).
$$

We introduce the following result about the Rademacher complexity.

**Proposition D.1** (From Theorem 2.1 in Bartlett et al. (2005))**.** *Let $\mathcal{F}$ be a class of functions that map $\mathcal{X}$ into $[a, b]$. Assume that there is some $r > 0$ such that for every $f \in \mathcal{F}$, $\mathrm{Var}(f(W)) \leq r$. Then, for every $z > 0$, with probability at least $1 - \exp(-z)$, it holds that*

$$
\sup_{f \in \mathcal{F}}\left(Pf - P_n f\right) \leq \inf_{\alpha > 0}\left\{2(1+\alpha)\mathbb{E}[\mathfrak{R}_n f] + \sqrt{\frac{2rx}{n}} + (b-a)\left(\frac{1}{3}+\frac{1}{\alpha}\right)\frac{z}{n}\right\}.
$$

### D.3   BRACKETING ENTROPY

We define the bracketing entropy. For a more detailed definition, see Definition 2.2 in van de Geer (2000).

**Definition D.1.** *Bracketing entropy. Given a class of functions $\mathcal{F}$, the logarithm of the smallest number of balls in a norm $\|\cdot\|_{2,P}$ of radius $\delta > 0$ needed to cover $\mathcal{F}$ is called the $\delta$-entropy with bracketing of $\mathcal{F}$ under the $L_2(P)$ metric, denoted by $H_B(\delta, \mathcal{F}, P)$.*

### D.4 TALAGRAND'S CONCENTRATION INEQUALITY

We introduce Talagrand's lemma.

**Proposition D.2** (Talagrand's Lemma). *Let $\phi\colon \mathbb{R} \to \mathbb{R}$ be a Lipschitz continuous function with a Lipschitz constant $L > 0$. Then, it holds that*

$$\mathfrak{R}_n(\phi \circ \mathcal{F}) \leq L\mathfrak{R}_n(\mathcal{F}).$$

## E BASIC INEQUALITIES

### E.1 STRONG CONVEXITY

**Lemma E.1** ($L_2$ distance bound from Lemma 4 in Kato & Teshima (2021)). *If $\inf_{h\in(-\infty),\infty} g''(h) > 0$, then there exists $\mu > 0$ such that for all $h \in \mathcal{H}$,*

$$\|h - h_0\|_2^2 \leq \frac{2}{\mu}\Big(\mathrm{BR}_g(h) - \mathrm{BR}_g(h_0)\Big)$$

*holds.*

From the strong convexity and Lemma E.1, we have

$$\frac{\mu}{2}\|\widehat{h}_n - h_0\|_2^2 \leq \mathrm{BR}_g(\widehat{h}_n) - \mathrm{BR}_g(h_0).$$

Recall that we have defined an estimator $\widehat{r}$ as follows:

$$\widehat{h} := \arg\min_{h\in\mathcal{H}} \widehat{\mathcal{L}}_n(h) + \lambda J(h),$$

where $J(h)$ is some regularization term.

### E.2 PRELIMINARY

**Proposition E.2.** *The estimator $\widehat{r}$ satisfies the following inequality:*

$$\widehat{\mathrm{BR}}_g(\widehat{h}) + \lambda J(\widehat{h}) \leq \widehat{\mathrm{BR}}_g(h^*) + \lambda J(h^*),$$

*where recall that*

$$\widehat{\mathrm{BR}}_g(h) := \frac{1}{n}\sum_{i=1}^n \Big( -g(h(D_i, X_i)) + \partial g(h(D_i, X_i))h(D_i, X_i) - \partial g(h(1, X_i)) - \partial g(h(0, X_i))\Big).$$

Let $Z \in \mathcal{Z}$ be a random variable with a space $\mathcal{Z}$, and $\{Z_i\}_{i=1}^n$ be its realizations. For a function $f\colon \mathcal{Z} \to \mathbb{R}$ and $X$ following $P$, let us denote the sample mean as

$$\widehat{\mathbb{E}}[f(Z)] := \frac{1}{n}\sum_{i=1}^n f(Z_i).$$

We also denote $\widehat{\mathbb{E}}[f(Z)] - \mathbb{E}[f(Z)] = (\widehat{\mathbb{E}} - \mathbb{E})f(Z)$

### E.3 RISK BOUND

Recall that

$$\widehat{\mathrm{BR}}_g(h) = \frac{1}{n}\sum_{i=1}^n \Big( -g(h(D_i, X_i)) + \partial g(h(D_i, X_i))h(D_i, X_i) - \partial g(h(1, X_i)) - \partial g(h(0, X_i))\Big).$$

Let us define

$$L(h, D, X) := -g(h(D, X)) + \partial g(h(D, X))h(D, X) - \partial g(h(1, X)) - \partial g(h(0, X)),$$

and we can write

$$\widehat{\mathrm{BR}}_g(h) = \widehat{\mathbb{E}}\big[L(h, D, X)\big]$$

Then, from Proposition E.2, we have

$$\widehat{\mathbb{E}}\big[L(h^*, D, X)\big] - \widehat{\mathbb{E}}\big[L(\widehat{h}_n, D, X)\big] + \lambda J(\widehat{h}) - \lambda J(h^*) \geq 0.$$

Throughout the proof, we use the following basic inequalities that hold for $\widehat{h}$.

**Proposition E.3.** *The estimator $\widehat{r}$ satisfies the following inequality:*

$$\frac{\mu}{2} \left\| \widehat{h}_n(D, X) - h_0(D, X) \right\|_{L_2(P_0)}^2$$

$$\leq \Big(\mathbb{E} - \widehat{\mathbb{E}}\Big) \Big[ L(\widehat{h}_n, D, X) - L(h_0, D, X) \Big] + \widehat{\mathbb{E}}\left[L(h^*, D, X) - L(h_0, D, X)\right] + \lambda J(r_0) - \lambda J(\widehat{r}).$$

Proof of Proposition E.2 is trivial. We prove Proposition E.3 below.

*Proof.* From the strong convexity and Lemma E.1, we have

$$\frac{\mu}{2} \|\widehat{h}_n - h_0\|_2^2 \leq \mathrm{BR}_g(\widehat{h}_n) - \mathrm{BR}_g(h_0) = \mathbb{E}\left[ L(\widehat{h}_n, D, X) - L(h_0, D, X) \right].$$

From Proposition E.2, we have

$$\frac{\mu}{2} \left\| \widehat{h}(D, X) - h_0(D, X) \right\|_{L_2(P_0)}^2$$

$$\leq \mathbb{E}\left[ L(\widehat{h}_n, D, X) - L(h_0, D, X) \right]$$

$$= \mathbb{E}\left[ L(\widehat{h}_n, D, X) - L(h_0, D, X) \right]$$

$$\quad - \widehat{\mathbb{E}}\left[ L(\widehat{h}_n, D, X) - L(h_0, D, X) \right]$$

$$\quad + \widehat{\mathbb{E}}\left[ L(\widehat{h}_n, D, X) - L(h_0, D, X) \right]$$

$$\leq \mathbb{E}\left[ L(\widehat{h}_n, D, X) - L(h_0, D, X) \right]$$

$$\quad - \widehat{\mathbb{E}}\left[ L(\widehat{h}_n, D, X) - L(h_0, D, X) \right]$$

$$\quad + \widehat{\mathbb{E}}\left[ L(\widehat{h}_n, D, X) - L(h_0, D, X) \right]$$

$$\quad - \widehat{\mathbb{E}}\left[ L(\widehat{h}_n, D, X) - L(h^*, D, X) \right] + \lambda J(\widehat{h}) - \lambda J(h_0).$$

$$\square$$

## F    PROOF OF THEOREM 4.1

We show Theorem 4.1 by bounding

$$\Big(\mathbb{E} - \widehat{\mathbb{E}}\Big) \Big[ L(\widehat{h}_n, D, X) - L(h_0, D, X) \Big], \tag{1}$$

in Proposition E.3. We can bound this term by using the empirical-process arguments.

Note that since $h_0 \in \mathcal{H}$, it holds that $h^* = h_0$, which implies that

### F.1    PRELIMINARY

We introduce the following propositions from van de Geer (2000), Kanamori et al. (2012) and Kato & Teshima (2021).

**Definition F.1** (Derived function class and bracketing entropy (from Definition 4 in Kato & Teshima (2021)))**.** *Given a real-valued function class $\mathcal{F}$, define $\ell \circ \mathcal{F} := \{\ell \circ f : f \in \mathcal{F}\}$. By extension, we define $I : \ell \circ \mathcal{H} \to [1, \infty)$ by $I(\ell \circ h) = I(h)$ and $\ell \circ \mathcal{H}_M := \{\ell \circ h : h \in \mathcal{H}_M\}$. Note that, as a result, $\ell \circ \mathcal{H}_M$ coincides with $\{\ell \circ h \in \ell \circ \mathcal{H} : I(\ell \circ h) \leq M\}$.*

**Proposition F.1.** *Let $\ell : \mathbb{R} \to \mathbb{R}$ be a $v$-Lipschitz continuous function. Let $H_B\big(\delta, \mathcal{F}, \|\cdot\|_{L_2(P_0)}\big)$ denote the bracketing entropy of $\mathcal{F}$ with respect to a distribution $P$. Then, for any distribution $P$, any $\gamma > 0$, any $M \geq 1$, and any $\delta > 0$, we have*

$$H_B\big(\delta, \ell \circ \mathcal{H}, \|\cdot\|_{L_2(P_0)}\big) \leq \frac{(s+1)(2v)^\gamma}{\gamma}\left(\frac{M}{\delta}\right)^\gamma.$$

*Moreover, there exists $M > 0$ such that for any $M \geq 1$ and any distribution $P$,*

$$\sup_{\ell \circ h \in \ell \circ \mathcal{H}_M} \|\ell \circ h - \ell \circ h^*\|_{L_2(P_0)} \leq c_0 v M,$$

$$\sup_{\substack{\ell \circ h \in \ell \circ \mathcal{H}_M \\ \|\ell \circ h - \ell \circ h^*\|_{L_2(P_0)} \leq \delta}} \|\ell \circ h - \ell \circ h^*\|_\infty \leq c_0 v M, \quad \text{for all } \delta > 0.$$

**Proposition F.2** (Lemma 5.13 in van de Geer (2000), Proposition 1 in Kanamori et al. (2012)). *Let $\mathcal{F} \subset L^2(P)$ be a function class and the map $I(f)$ be a complexity measure of $f \in \mathcal{F}$, where $I$ is a non-negative function on $\mathcal{F}$ and $I(f_0) < \infty$ for a fixed $f_0 \in \mathcal{F}$. We now define $\mathcal{F}_M = \{f \in \mathcal{F} : I(f) \leq M\}$ satisfying $\mathcal{F} = \bigcup_{M \geq 1} \mathcal{F}_M$. Suppose that there exist $c_0 > 0$ and $0 < \gamma < 2$ such that*

$$\sup_{f \in \mathcal{F}_M} \|f - f_0\| \leq c_0 M, \qquad \sup_{\substack{f \in \mathcal{F}_M \\ \|f - f_0\|_{L^2(P)} \leq \delta}} \|f - f_0\|_\infty \leq c_0 M, \quad \text{for all } \delta > 0,$$

*and that $H_B(\delta, \mathcal{F}_M, P) = O\left((M/\delta)^\gamma\right)$. Then, we have*

$$\sup_{f \in \mathcal{F}} \frac{\left|\int(f - f_0)d(P - P_n)\right|}{D(f)} = O_p(1), \ (n \to \infty),$$

*where $D(f)$ is defined by*

$$D(f) = \max \frac{\|f - f_0\|_{L^2(P)}^{1-\gamma/2} I(f)^{\gamma/2}}{\sqrt{n}} \frac{I(f)}{n^{2/(2+\gamma)}}.$$

**Proposition F.3.** *Let $g \colon \mathcal{K} \to \mathbb{R}$ be twice continuously differentiable and strictly convex for the space $\mathcal{K}$ of $h_0$, and suppose that there exists $M > 0$ such that*

$$|g''(t)| \leq M \quad \text{for all } t \in \mathbb{R}.$$

*Let $\zeta^{-1} \colon \mathbb{R} \to \mathbb{R}$ be continuously differentiable and globally Lipschitz, that is, there exists $L_\zeta > 0$ such that*

$$|\zeta^{-1}(s) - \zeta^{-1}(t)| \leq L_\zeta |s - t| \quad \text{for all } s, t \in \mathbb{R}.$$

*Assume also that $\zeta^{-1}(0)$ is finite, and define*

$$a_0 := |\zeta^{-1}(0)|, \qquad a_1 := L_\zeta,$$

*so that*

$$|\zeta^{-1}(u)| \leq a_0 + a_1|u| \quad \text{for all } u \in \mathbb{R}.$$

*Let $h$ be a bounded real-valued function on the domain of $(D, X)$, and write*

$$\|h\|_\infty := \sup_{d,x} |h(d, x)|.$$

*Let $L$ be a linear functional acting on bounded functions, such that for some constant $C_L > 0$,*

$$|L(f)| \leq C_L\big(1 + \|f\|_\infty\big) \quad \text{for all bounded } f.$$

*Define*

$$L(\zeta^{-1} \circ f) = g\big(\zeta^{-1} \circ f(D, X)\big) + \partial g\big(\zeta^{-1} \circ f(D, X)\big)\zeta^{-1} \circ h(D, X)$$
$$- \partial g\big(\zeta^{-1} \circ f(1, X)\big) - \partial g\big(\zeta^{-1} \circ f(0, X)\big).$$

*Then there exists a constant $C > 0$ (depending only on $g$, $\zeta^{-1}$ and $C_L$) such that*

$$|L(\zeta^{-1} \circ f)| \leq C\big(1 + \|f\|_\infty^2\big).$$

### F.2 UPPER BOUND USING THE EMPIRICAL-PROCESS ARGUMENTS

From Propositions F.1–F.3, we obtain the following result.

**Proposition F.4.** *Under the conditions of Theorem 4.1, for any $0 < \gamma < 2$, we have*

$$d\left(\mathbb{E} - \widehat{\mathbb{E}}\right)\left[L(\widehat{h}_n, D, X) - L(h_0, D, X)\right]$$

$$= O_p\left(\max\left\{\frac{\|\widehat{h}_n - h^*\|_{L^2(P_0)}^{1-\gamma/2}\left(1 + \left\|\widehat{h}_n\right\|_{\mathcal{H}}\right)^{1+\gamma/2}}{\sqrt{n}}, \frac{\left(1 + \left\|\widehat{h}_n\right\|_{\mathcal{H}}\right)^2}{n^{2/(2+\gamma)}}\right\}\right),$$

*as $n \to \infty$.*

### F.3 PROOF OF THEOREM 4.1

We prove Theorem 4.1 following the arguments in Kanamori et al. (2012).

*Proof.* From Proposition E.3 and $h_0 \in \mathcal{H}^{\mathrm{RKHS}}$, we have

$$\left\|\widehat{h}_n(D, X) - h_0(D, X)\right\|_{L_2(P_0)}^2 + \lambda\|\widehat{h}\|_{\mathcal{H}}^2$$

$$\leq \left(\mathbb{E} - \widehat{\mathbb{E}}\right)\left[L(\widehat{h}_n, D, X) - L(h_0, D, X)\right] + \lambda\|f_0\|_{\mathcal{H}}^2.$$

From Proposition F.4, we have

$$\left\|\widehat{h}_n(D, X) - h_0(D, X)\right\|_{L_2(P_0)}^2 + \lambda\|\widehat{f}_n\|_{\mathcal{H}}^2$$

$$= \emptyset_p\left(\max\left\{\frac{\|\widehat{h} - h_0\|_{L^2(P_0)}^{1-\gamma/2}\left(1 + \left\|\widehat{f}\right\|_{\mathcal{H}}\right)^{1+\gamma/2}}{\sqrt{n}}, \frac{\left(1 + \left\|\widehat{h}\right\|_{\mathcal{H}}\right)^2}{n^{2/(2+\gamma)}}\right\}\right) + \lambda\|r_0\|_{\mathcal{H}}^2.$$

We consider the following three possibilities:

$$\left\|\widehat{h}_n(D, X) - h_0(D, X)\right\|_{L_2(P_0)}^2 + \lambda\|\widehat{f}_n\|_{\mathcal{H}}^2 = O_p(\lambda), \tag{2}$$

$$\left\|\widehat{h}_n(D, X) - h_0(D, X)\right\|_{L_2(P_0)}^2 + \lambda\|\widehat{f}_n\|_{\mathcal{H}}^2 = O_p\left(\frac{\|\widehat{f} - f_0\|_{L^2(P_0)}^{1-\gamma/2}\left(1 + \left\|\widehat{f}\right\|_{\mathcal{H}}\right)^{1+\gamma/2}}{\sqrt{n}}\right), \tag{3}$$

$$\left\|\widehat{h}_n(D, X) - h_0(D, X)\right\|_{L_2(P_0)}^2 + \lambda\|\widehat{f}_n\|_{\mathcal{H}}^2 = O_p\left(\frac{\left(1 + \left\|\widehat{f}\right\|_{\mathcal{H}}\right)^2}{n^{2/(2+\gamma)}}\right). \tag{4}$$

The above inequalities are analyzed as follows:

**Case (2).** We have

$$\left\|\widehat{h}_n(D, X) - h_0(D, X)\right\|_{L_2(P_0)}^2 = O_p(\lambda),$$

$$\lambda\|\widehat{f}_n\|_{\mathcal{H}}^2 = O_p(\lambda).$$

Therefore, we have $\left\|\widehat{h}_n(D, X) - h_0(D, X)\right\|_{P_0} = O_p(\lambda^{1/2})$ and $\|\widehat{r}\|_{\mathcal{H}} = O_p(1)$.

**Case (3).** We have

$$\left\|\widehat{h}_n(D,X) - h_0(D,X)\right\|^2_{L_2(P_0)} = O_p\left(\frac{\|\widehat{f}_n - f_0)\|^{1-\gamma/2}_{L^2(P_0)}\left(1 + \left\|\widehat{f}_n\right\|_{\mathcal{F}}\right)^{1+\gamma/2}}{\sqrt{n}}\right),$$

$$\lambda\|\widehat{f}_n\|^2_{\mathcal{H}} = O_p\left(\frac{\|\widehat{f}_n - f_0)\|^{1-\gamma/2}_{L^2(P_0)}\left(1 + \left\|\widehat{f}_n\right\|_{\mathcal{F}}\right)^{1+\gamma/2}}{\sqrt{n}}\right).$$

From the first inequality, we have

$$\left\|\widehat{h}_n(D,X) - h_0(D,X)\right\|_{P_0} = \sum_{d\in\{1,0\}} O_p\left(\frac{\left(1 + \left\|\widehat{f}_n\right\|_{\mathcal{F}}\right)^{1+\gamma/2}}{n^{1/(2+\gamma)}}\right).$$

By using this result, from the second inequality, we have

$$\lambda\|\widehat{f}_n\|^2_{\mathcal{H}} = O_p\left(\frac{\|\widehat{f}_n - f_0)\|^{1-\gamma/2}_{L^2(P_0)}\left(1 + \left\|\widehat{f}_n\right\|_{\mathcal{F}}\right)^{1+\gamma/2}}{\sqrt{n}}\right)$$

$$= O_p\left(\left(\frac{1 + \left\|\widehat{f}_n\right\|_{\mathcal{F}}}{n^{1/(2+\gamma)}}\right)^{1-\gamma/2}\frac{\left(1 + \left\|\widehat{f}_n\right\|_{\mathcal{F}}\right)^{1+\gamma/2}}{\sqrt{n}}\right)$$

$$= O_p\left(\frac{\left(1 + \left\|\widehat{f}_n\right\|_{\mathcal{F}}\right)^2}{n^{2/(2+\gamma)}}\right).$$

This implies that

$$\|\widehat{f}\|_{\mathcal{H}} = O_p\left(\frac{\left(1 + \left\|\widehat{f}_n\right\|_{\mathcal{F}}\right)^2}{\lambda^{1/2}n^{2/(2+\gamma)}}\right) = o_p(1).$$

Therefore, the following inequity is obtained.

$$\left\|\widehat{h}_n(D,X) - h_0(D,X)\right\|_{P_0} = O_p\left(\frac{1}{n^{1/(2+\gamma)}}\right) = O_p(\lambda^{1/2}).$$

**Case 4.** We have

$$\left\|\widehat{h}_n(D,X) - h_0(D,X)\right\|^2_{L_2(P_0)} = O_p\left(\frac{\left(1 + \left\|\widehat{f}_n\right\|_{\mathcal{F}}\right)^2}{n^{2/(2+\gamma)}}\right),$$

$$\lambda\|\widehat{f}_n\|^2_{\mathcal{H}} = O_p\left(\frac{\left(1 + \left\|\widehat{f}_n\right\|_{\mathcal{F}}\right)^2}{n^{2/(2+\gamma)}}\right).$$

As well as the argument in (3), we have $\|\widehat{r}\|_{\mathcal{H}} = o_p(1)$. Therefore, we have

$$\left\|\widehat{h}_n(D,X) - h_0(D,X)\right\|_{P_0} = O_p\left(\frac{1}{n^{1/(2+\gamma)}}\right) = O_p(\lambda^{1/2}).$$

$\square$

# G   PROOF OF THEOREM 4.2

Our proof procedure mainly follows those in Kato & Teshima (2021) and Zheng et al. (2022). In particular, we are inspired by the proof in Zheng et al. (2022).

We prove Theorem 4.2 by proving the following lemma:

**Lemma G.1.** *Suppose that Assumption 4.2 holds. For any $n \geq \mathrm{Pdim}(\mathcal{F}^{\mathrm{FNN}})$, there exists a constant $C > 0$ depending on $(\mu, \sigma, M)$ such that for any $\gamma > 0$, with probability at least $1 - \exp(-\gamma)$, it holds that*

$$\left\| \widehat{f}_n - f_0 \right\|_2 \leq C \left( \sqrt{\frac{\mathrm{Pdim}(\mathcal{F}^{\mathrm{FNN}}) \log(n)}{n}} + \left\| f^* - f_0 \right\|_2 + \sqrt{\frac{\gamma}{n}} \right).$$

As shown in Zheng et al. (2022), we can bound $\mathrm{Pdim}(\mathcal{F}^{\mathrm{FNN}}) \log(n)$ by specifying neural networks and obtain Theorem 4.2.

## G.1   PROOF OF LEMMA G.1

We prove Lemma G.1 by bounding (1) in Proposition E.3.

To bound (1), we show several auxiliary results. Define

$$\widehat{\mathcal{F}}^{f^*, u} := \{ f \in \mathcal{F}^{\mathrm{FNN}} \colon \frac{1}{n} \sum_{i=1}^{n} (f(D_i, X_i) - f^*(D_i, X_i))^2 \leq u \},$$

$$\overline{\mathcal{G}}^{f^*, u} := \left\{ (f - f^*) \colon f \in \widehat{\mathcal{F}}^{f^*, u} \right\},$$

$$\kappa_n^u(u) := \mathbb{E}_\sigma \left[ \mathfrak{R}_n \overline{\mathcal{G}}^{f^*, u} \right],$$

$$u^\dagger := \inf \left\{ u \geq 0 \colon \kappa_n^u(s) \leq s^2 \quad \forall s \geq u \right\}.$$

Here, we show the following two lemmas:

**Lemma G.2** (Corresponding to (26) in Zheng et al. (2022))**.** *Suppose that the conditions in Lemma G.1 hold. Then, for any $z > 0$, with probability $1 - \exp(-z)$ it holds that*

$$\widehat{\mathbb{E}} \left[ L(\widehat{h}_n, D, X) - L(h_0, D, X) \right]$$

$$\leq C \left( \| f^*(D, X) - f_0(D, X) \|_2^2 + \| f^*(D, X) - f_0(D, X) \|_2 \sqrt{\frac{z}{n}} + \frac{16 M z}{3n} \right).$$

**Lemma G.3** (Corresponding to (29) in Zheng et al. (2022))**.** *Suppose that the conditions in Lemma G.1 hold. If there exists $u_0 > 0$ such that*

$$\| \widehat{f}(D, X) - f^*(D, X) \|_2 \leq u_0,$$

*then it holds that*

$$\left( \mathbb{E} - \widehat{\mathbb{E}} \right) \left[ L(\widehat{h}_n, D, X) - L(h_0, D, X) \right]$$

$$\leq C \left( \mathbb{E}_\sigma \left[ \mathfrak{R}_n \overline{\mathcal{G}}^{f^*, u_0} \right] + u_0 \sqrt{\frac{z}{n}} + \frac{M z}{n} \right).$$

Additionally, we use the following three propositions directly from Zheng et al. (2022).

**Proposition G.4** (From (32) in Zheng et al. (2022))**.** *Let $u > 0$ be a positive value such that*

$$\| f - f_0 \|_2 \leq u$$

*for all $f \in \mathcal{F}$. Then, for every $z > 0$, with probability at least $1 - 2 \exp(-z)$, it holds that*

$$\sqrt{\frac{1}{n} \sum_{i=1}^{n} \left( f(X_i) - f_0(X_i) \right)^2} \leq 2u.$$

**Proposition G.5** (Corresponding to (36) in Step 3 of Zheng et al. (2022)). *Suppose that the conditions in Lemma G.1 hold. Then, there exists a universal constant $C > 0$ such that*

$$u^\dagger \leq CM\sqrt{\frac{\text{Pdim}(\mathcal{F}^{\text{FNN}})\log(n)}{n}}.$$

**Proposition G.6** (Upper bound of the Rademacher complexity). *Suppose that the conditions in Lemma G.1 hold. If $n \geq \text{Pdim}(\mathcal{F}^{\text{FNN}})$, $u_0 \geq 1/n$, and $n \geq (2eM)^2$, we have*

$$\mathbb{E}_\sigma\left[\mathfrak{R}_n\overline{\mathcal{G}}^{f^*,u_0}\right] \leq Cr_0\sqrt{\frac{\text{Pdim}(\mathcal{F}^{\text{FNN}})\log n}{n}}.$$

Then, we prove Lemma G.1 as follows:

*Proof of Lemma G.1.* If there exists $u_0 > 0$ such that

$$\|\widehat{f}(X) - f^*(X)\|_2 \leq u_0,$$

then from (1) and Lemmas G.2 and G.3, for every $z > 0$, there exists a constant $C > 0$ independent $n$ such that

$$\left\|\widehat{h}_n(D, X) - h_0(D, X)\right\|^2_{L_2(P_0)}$$

$$\leq C\left(\|f^* - f_0\|_2\sqrt{\frac{z}{n}} + \frac{16Mz}{3n} + u_0\sqrt{\frac{\text{Pdim}(\mathcal{F}^{\text{FNN}})\log n}{n}} + u_0\sqrt{\frac{z}{n}} + \frac{Mz}{n}\right). \tag{5}$$

This result implies that if $\sqrt{\text{Pdim}(\mathcal{F}^{\text{FNN}})}$, then there exists $n_0$ such that for all $n > n_0$, there exists $u_1 < u_0$ such that

$$\left\|\widehat{h}_n(D, X) - h_0(D, X)\right\|^2_{L_2(P_0)} \leq u_1.$$

For any $z > 0$, define $\overline{u}$ as

$$\overline{u}_z \geq \max\left\{\sqrt{\log(n)/n}, 4\sqrt{3}M\sqrt{z/n}, u^\dagger\right\}.$$

Define a subspace of $\mathcal{F}^{\text{FNN}}$ as

$$\mathcal{S}^{\text{FNN}}(f_0, \overline{u}_z := \left\{f \in \mathcal{F}^{\text{FNN}} : \|f - f_0\| \leq \overline{u}_z\right\}.$$

Define

$$\ell := \lfloor\log_2(2M/\sqrt{\log(n)/n})\rfloor.$$

Using the definition of subspaces, we divide $\mathcal{F}^{\text{FNN}}$ into the following $\ell + 1$ subspaces:

$$\overline{\mathcal{S}}^{\text{FNN}}_0 := \mathcal{S}^{\text{FNN}}(f_0, \overline{u}),$$
$$\overline{\mathcal{S}}^{\text{FNN}}_1 := \mathcal{S}^{\text{FNN}}(f_0, \overline{u})\backslash\mathcal{S}^{\text{FNN}}(f_0, \overline{u}),$$
$$\vdots$$
$$\overline{\mathcal{S}}^{\text{FNN}}_\ell := \mathcal{S}^{\text{FNN}}(f_0, 2^\ell\overline{u})\backslash\mathcal{S}^{\text{FNN}}(f_0, 2^{\ell-1}\overline{u}).$$

Since $\overline{u}_z > u^\dagger$, from the definition of $u^\dagger$, we have

$$\overline{u}_z^2 \leq \kappa_n^u(\overline{u}).$$

If there exists $j \leq \ell$ such that $\widehat{f} \in \overline{\mathcal{S}}^{\text{FNN}}_j$, then from (5), for every $z > 0$, with probability at least $1 - 8\exp(-z)$, there exists a constant $C > 0$ independent of $n$ such that

$$\left\|\widehat{h}_n(D, X) - h_0(D, X)\right\|^2_2$$

$$\leq C \left( 2^{\ell-1}\overline{u} \left( \sqrt{\frac{\mathrm{Pdim}(\mathcal{F}^{\mathrm{FNN}})\log(n)}{n}} + \sqrt{\frac{z}{n}} \right) + \|f^* - f_0\|_2^2 + \|f^* - f_0\|_2\sqrt{\frac{z}{n}} + \frac{Mz}{n} \right). \tag{6}$$

Additionally, if

$$C \left( \sqrt{\frac{\mathrm{Pdim}(\mathcal{F}^{\mathrm{FNN}})\log(n)}{n}} + \sqrt{\frac{z}{n}} \right) \leq \frac{1}{8}2^j\overline{u}, \tag{7}$$

$$C \left( \|f^* - f_0\|_2^2 + \|f^* - f_0\|_2\sqrt{\frac{z}{n}} + \frac{Mz}{n} \right) \leq \frac{1}{8}2^{2j}\overline{u}^2 \tag{8}$$

hold, then

$$\left\| \widehat{h}_n(D, X) - h_0(D, X) \right\|_2 \leq 2^{j-1}\overline{u}. \tag{9}$$

Here, to obtain (9), we used $\overline{u} \geq \max\left\{ \sqrt{\log(n)/n}, 4\sqrt{3}M\sqrt{z/n}, u^\dagger \right\}$, (6), (7), and (8).

From Proposition G.5, it holds that

$$u^\dagger \leq CM\sqrt{\frac{\mathrm{Pdim}(\mathcal{F}^{\mathrm{FNN}})\log(n)}{n}}.$$

Therefore, we can choose $\overline{u}$ as

$$\overline{u} := C \left( \sqrt{\frac{\mathrm{Pdim}(\mathcal{F}^{\mathrm{FNN}})\log(n)}{n}} + \sqrt{\log(n)/n} + 4\sqrt{3}M\sqrt{z/n} \right),$$

where $C > 0$ is a constant independent of $n$. $\qquad\square$

## G.2 PROOF OF LEMMA G.2

From Proposition D.1, we have

$$\widehat{\mathbb{E}}\left[ L(\widehat{h}_n, D, X) - L(h_0, D, X) \right]$$

$$\leq \mathbb{E}\left[ L(\widehat{h}_n, D, X) - L(h_0, D, X) \right] + \sqrt{2}C\|f^*(X) - f_0(X)\|\sqrt{\frac{z}{n}} + \frac{16C_1Mz}{3n}.$$

This is a direct consequence of Proposition D.1. Note that $h^*$ and $h_0$ are fixed, and it is enough to apply the standard law of large numbers; that is, we do not have to consider the uniform law of large numbers. However, we can still apply Proposition D.1, which is a general than the standard law of large numbers, with ignoring the Rademacher complexity part.

We have

$$\widehat{\mathbb{E}}\left[ L(\widehat{h}_n, D, X) - L(h_0, D, X) \right]$$

$$\leq \mathbb{E}\left[ L(\widehat{h}_n, D, X) - L(h_0, D, X) \right]$$

$$+ \sqrt{2}C_1\|f^* - f_0\|\sqrt{\frac{z}{n}} + \frac{16C_2Mz}{3n} + \sqrt{2}C_2\|f^* - f_0\|\sqrt{\frac{z}{n}} + \frac{16C_2Mz}{3n}$$

$$\leq C \left( \|f^* - f_0\|_2^2 + \|f^* - f_0\|\sqrt{\frac{z}{n}} + \frac{16CMz}{3n} \right).$$

## G.3 PROOF OF LEMMA G.3

Let $g := (f - f^*)^2$. From the definition of FNNs, we have

$$g \leq 4M^2$$

Additionally, we assumed that $\|\widehat{f} - f^*\|_2 \leq u_0$ holds. Then, it holds that $\mathrm{Var}_{P_0}(g) \leq 4M^2u_0^2$.

Here, we note that the followings hold for all $f(r)$:

$$L(h) - L(h^*) \le C\Big|f(d,x) - f^*(d,x)\Big|,$$

where $C > 0$ is some constant

Then, from Proposition D.1, for every $z > 0$, with probability at least $1 - \exp(-z)$, it holds that

$$\left(\mathbb{E} - \widehat{\mathbb{E}}\right)\left[L(\widehat{h}_n, D, X) - L(h_0, D, X)\right]$$
$$\le C\left(\mathbb{E}_\sigma\left[\mathfrak{R}_n \overline{\mathcal{G}}^{f^*, u_0}\right] + r_0\sqrt{\frac{z}{n}} + \frac{Mz}{n}\right).$$

## H  NEAREST NEIGHBOR MATCHING

In this section, we show that nearest neighbor (NN) matching for the ATE can be interpreted as a special case of our direct bias-correction term estimation with the squared loss, that is, Riesz regression or LSIF. This result is shown in Kato (2025b), a subsequent work of this study.

The key step is to express the ATE bias-correction term $h_0(D, X)$ in terms of density ratios with respect to the marginal covariate distribution and then to approximate these density ratios via nearest neighbor cells, following the density-ratio interpretation in Lin et al. (2023).

### H.1  ATE BIAS-CORRECTION TERM AND DENSITY RATIOS

Let $p_X$ denote the marginal density of $X$ and $p_{X|D=d}$ the conditional density of $X$ given $D = d$. Let $\pi_1 := P_0(D = 1)$ and $\pi_0 := P_0(D = 0) = 1 - \pi_1$. By Bayes' rule,

$$p_{X|D=d}(x) = \frac{p_X(x)P_0(D = d \mid X = x)}{P_0(D = d)} = \frac{p_X(x)e_0(x)^d(1 - e_0(x))^{1-d}}{\pi_d},$$

where $\pi_d = P_0(D = d)$ and $e_0(x) = P_0(D = 1 \mid X = x)$.

Define the density ratios with respect to the marginal distribution of $X$ by

$$r_1(x) := \frac{p_X(x)}{p_{X|D=1}(x)}, \qquad r_0(x) := \frac{p_X(x)}{p_{X|D=0}(x)}.$$

From the expression above,

$$r_1(x) = \frac{\pi_1}{e_0(x)}, \qquad r_0(x) = \frac{\pi_0}{1 - e_0(x)}.$$

Therefore, the ATE bias-correction term

$$h_0(D, X) = \frac{\mathbb{1}[D = 1]}{e_0(X)} - \frac{\mathbb{1}[D = 0]}{1 - e_0(X)}$$

can be written in terms of $r_1$ and $r_0$ as

$$h_0(D, X) = \mathbb{1}[D = 1]\frac{r_1(X)}{\pi_1} - \mathbb{1}[D = 0]\frac{r_0(X)}{\pi_0}. \tag{10}$$

Thus, estimating $h_0$ is equivalent to estimating the pair $(r_1, r_0)$, the density ratios between the marginal covariate distribution and the treated and control covariate distributions.

### H.2  SQUARED LOSS OBJECTIVE AND DECOMPOSITION INTO TWO LSIF PROBLEMS

Recall that when we choose the squared loss $g^{\mathrm{SL}}(h) = (h - 1)^2$, the population Bregman divergence objective for $h$ is

$$\mathrm{BR}_{g^{\mathrm{SL}}}(h) = \mathbb{E}\Big[-2\big(h(1, X) - h(0, X)\big) + h(D, X)^2\Big].$$

Consider the parameterization

$$h(D, X) = \mathbb{1}[D = 1]\frac{r_1(X)}{\pi_1} - \mathbb{1}[D = 0]\frac{r_0(X)}{\pi_0},$$

with $r_1, r_0$ defined above. Substituting this into $\mathrm{BR}_{g^{\mathrm{SL}}}(h)$ and using the law of total expectation, we obtain

$$\mathrm{BR}_{g^{\mathrm{SL}}}(h) = C - 2\mathbb{E}\Big[\frac{r_1(X)}{\pi_1} + \frac{r_0(X)}{\pi_0}\Big] + \mathbb{E}\Big[h(D,X)^2\Big], \tag{11}$$

where $C$ is a constant independent of $(r_1, r_0)$. The last term can be decomposed as

$$\mathbb{E}\big[h(D,X)^2\big] = \pi_1 \mathbb{E}\Big[\Big(\frac{r_1(X)}{\pi_1}\Big)^2 \,\Big|\, D=1\Big] + \pi_0 \mathbb{E}\Big[\Big(\frac{r_0(X)}{\pi_0}\Big)^2 \,\Big|\, D=0\Big].$$

Rewriting (11) in terms of expectations with respect to $p_X$ and $p_{X|D=d}$ and dropping constants gives

$$\mathrm{BR}_{g^{\mathrm{SL}}}(h) := -2\mathbb{E}_X\left[r_1(X)\right] + \mathbb{E}_{X|D=1}\left[r_1(X)^2\right] - 2\mathbb{E}_X\left[r_0(X)\right] + \mathbb{E}_{X|D=0}\left[r_0(X)^2\right]. \tag{12}$$

Hence minimizing $\mathrm{BR}_{g^{\mathrm{SL}}}(h)$ over $(r_1, r_0)$ is equivalent to solving two independent LSIF-type problems

$$r_1^* = \arg\min_{r_1} \left\{ -2\mathbb{E}_X[r_1(X)] + \mathbb{E}_{X|D=1}[r_1(X)^2] \right\},$$

$$r_0^* = \arg\min_{r_0} \left\{ -2\mathbb{E}_X[r_0(X)] + \mathbb{E}_{X|D=0}[r_0(X)^2] \right\},$$

and then plugging $(r_1^*, r_0^*)$ into (10).

At the sample level, with $\mathcal{G}_1$ and $\mathcal{G}_0$ defined as in the Introduction, the empirical LSIF objectives are

$$\widehat{J}_1(r_1) := -\frac{2}{n}\sum_{i=1}^n r_1(X_i) + \frac{1}{|\mathcal{G}_1|}\sum_{i\in\mathcal{G}_1} r_1(X_i)^2, \tag{13}$$

$$\widehat{J}_0(r_0) := -\frac{2}{n}\sum_{i=1}^n r_0(X_i) + \frac{1}{|\mathcal{G}_0|}\sum_{i\in\mathcal{G}_0} r_0(X_i)^2. \tag{14}$$

Minimizing $\widehat{J}_1$ and $\widehat{J}_0$ and then using (10) yields an LSIF (Riesz regression) estimator of the ATE bias-correction term $h_0$.

### H.3 Nearest-neighbor partition and histogram model

To connect this LSIF formulation to nearest neighbor matching, we now choose a simple histogram-type model for $(r_1, r_0)$ based on nearest neighbor cells. Let us consider the $M$-nearest neighbor partition induced by the sample $\{X_i\}_{i=1}^n$.

For each treated unit $i \in \mathcal{G}_1$, let $N_M^{(0)}(i) \subset \mathcal{G}_0$ denote the set of $M$ nearest control units to $X_i$. Similarly, for each control unit $j \in \mathcal{G}_0$, let $N_M^{(1)}(j) \subset \mathcal{G}_1$ denote the set of $M$ nearest treated units to $X_j$. We define the neighbor counts

$$K_M^{(1)}(k) := \big|\{i \in \mathcal{G}_1 : k \in N_M^{(0)}(i)\}\big|, \qquad K_M^{(0)}(k) := \big|\{j \in \mathcal{G}_0 : k \in N_M^{(1)}(j)\}\big|.$$

Thus $K_M^{(1)}(k)$ counts how often unit $k$ is selected as a control neighbor of treated units, and $K_M^{(0)}(k)$ counts how often it is selected as a treated neighbor of control units. The total numbers of neighbor links are

$$\sum_{k=1}^n K_M^{(1)}(k) = M|\mathcal{G}_1|, \qquad \sum_{k=1}^n K_M^{(0)}(k) = M|\mathcal{G}_0|.$$

We now approximate each density ratio $r_d$ by a histogram that is constant on the Voronoi cells induced by the sample:

$$r_d(x) = \sum_{k=1}^n \theta_k^{(d)}\psi_k(x),$$

where $\{\psi_k\}_{k=1}^n$ is the partition of $\mathcal{X}$ such that $\psi_k(x) = 1$ if $x$ lies in the cell associated with $X_k$ and $\psi_k(x) = 0$ otherwise. Approximating the integrals in (13) and (14) by assigning each observation $X_i$ to the nearest cell, the empirical objectives become (up to constants)

$$\widehat{J}_1(\theta^{(1)}) \approx -\frac{2}{n}\sum_{k=1}^n K_M^{(X)}(k)\,\theta_k^{(1)} + \frac{1}{|\mathcal{G}_1|}\sum_{k=1}^n \mathbb{1}[k \in \mathcal{G}_1]\big(\theta_k^{(1)}\big)^2, \tag{15}$$

Table 3: Results of additional simulation studies. CR denotes the coverage ratio of 95% confidence intervals; that is, values close to $0.95$ are better. DM denotes the direct method, which is independent of the direct bias-correction term estimation methods; therefore, in theory, the results of the DM estimator should not differ across DBC (LS), DBC (KL), and DBC (TL). Since we compute the DM estimator when constructing the AIPW estimator in each of DBC (LS), DBC (KL), and DBC (TL), we also report the DM estimator results for reference.

| | True | | | DBC (LS) | | | DBC (KL) | | | DBC (TL) | | |
|---|---|---|---|---|---|---|---|---|---|---|---|---|
| | DM | IPW | AIPW | DM | IPW | AIPW | DM | IPW | AIPW | DM | IPW | AIPW |
| MSE | 0.00 | 1.10 | 0.01 | 0.30 | 0.59 | 0.11 | 0.30 | 0.41 | 0.08 | 0.31 | 0.36 | 0.09 |
| CR | 1.00 | 0.92 | 0.97 | 0.17 | 0.97 | 0.87 | 0.11 | 0.97 | 0.88 | 0.11 | 0.92 | 0.87 |

$$\widehat{J}_0(\theta^{(0)}) \approx -\frac{2}{n}\sum_{k=1}^{n} K_M^{(X)}(k)\,\theta_k^{(0)} + \frac{1}{|\mathcal{G}_0|}\sum_{k=1}^{n}\mathbb{1}[k \in \mathcal{G}_0]\big(\theta_k^{(0)}\big)^2, \tag{16}$$

where $K_M^{(X)}(k)$ denotes the number of times $X_k$ is selected as a nearest neighbor when we run the $M$-NN search over the whole sample $\{X_i\}_{i=1}^{n}$.[2]

Minimizing the quadratic objectives (15) and (16) with respect to each $\theta_k^{(d)}$ yields the closed-form solutions

$$\theta_k^{(1)*} \propto K_M^{(X)}(k)\mathbb{1}[k \in \mathcal{G}_1], \qquad \theta_k^{(0)*} \propto K_M^{(X)}(k)\mathbb{1}[k \in \mathcal{G}_0].$$

Therefore, up to a common normalization constant,

$$r_1(X_k) \propto K_M^{(X)}(k)\mathbb{1}[k \in \mathcal{G}_1], \qquad r_0(X_k) \propto K_M^{(X)}(k)\mathbb{1}[k \in \mathcal{G}_0].$$

Substituting these expressions into (10) gives

$$h^{\mathrm{NN}}(D_k, X_k) = (2D_k - 1)\Big(1 + \frac{K_M^{(X)}(k)}{M}\Big) \times c_n, \tag{17}$$

for some sample-size dependent normalization constant $c_n$. Equation (17) coincides, up to normalization, with the nearest-neighbor based bias-correction weights derived in Lin et al. (2023) for the ATE.

### H.4 NEAREST NEIGHBOR MATCHING AS RIESZ REGRESSION

Using the bias-correction term $h^{\mathrm{NN}}$ in (17), the corresponding IPW-type ATE estimator becomes

$$\widehat{\tau}^{\mathrm{NN}} = \frac{1}{n}\sum_{k=1}^{n} h^{\mathrm{NN}}(D_k, X_k)Y_k,$$

which can be expanded to the familiar $M$-nearest neighbor matching form

$$\widehat{\tau}^{\mathrm{NN}} = \frac{1}{n}\sum_{i\in\mathcal{G}_1}\Big(Y_i - \frac{1}{M}\sum_{j\in N_M^{(0)}(i)} Y_j\Big) - \frac{1}{n}\sum_{j\in\mathcal{G}_0}\Big(Y_j - \frac{1}{M}\sum_{i\in N_M^{(1)}(j)} Y_i\Big),$$

that is, a two-sided nearest neighbor matching estimator for the ATE that matches treated units to control units and control units to treated units. Therefore, nearest neighbor matching for the ATE is obtained by minimizing the squared-loss Bregman divergence within a nearest-neighbor histogram model for the density ratios $(r_1, r_0)$ and then plugging the resulting estimator into the bias-correction term $h(D, X)$.

In other words, nearest neighbor matching is a special case of Riesz regression (LSIF) with a particular choice of feature dictionary based on nearest neighbor cells. This formally justifies the statement in the main text that nearest neighbor matching can be interpreted as a direct bias-correction term estimator obtained from our squared-loss Bregman divergence framework.

## I ADDITIONAL SIMULATION STUDIES

In this section, we conduct additional simulation studies to more closely examine the finite sample behavior of our direct bias-correction approach under different choices of Bregman divergence. We focus on the three representative losses

---

[2]For a detailed derivation of this approximation, see the analysis of histogram LSIF in Lin et al. (2023).

introduced in Section 2: the squared loss corresponding to Riesz regression (denoted by DBC (LS)), the KL divergence loss (DBC (KL)), and the tailored loss (DBC (TL)). We refer to our method collectively as the direct bias-correction (DBC) approach.

Unlike the simulation design in Section 2 (Simulation studies), here we explicitly use cross fitting in the sense of Assumption 5.1. This setting illustrates how our framework can be combined with modern high-capacity models without requiring the Donsker assumption.

## I.1   DESIGN AND IMPLEMENTATION

We consider the same basic ATE setting as in the previous simulations. The covariates are three dimensional, $K = 3$, and we fix the sample size at $n = 3000$. In each Monte Carlo replication, we generate covariates $X_i \in \mathbb{R}^3$ from a multivariate normal distribution $\mathcal{N}(0, I_3)$, and construct a nonlinear propensity score model with polynomial and interaction terms, as in the main simulation study. Treatment assignments $D_i$ are then sampled from the resulting Bernoulli distribution with success probability $e_0(X_i)$. The outcome $Y_i$ is generated from a nonlinear regression model that includes both squared terms and a nonlinear transformation, with the true ATE fixed at $\tau_0 = 5.0$. The noise term is standard normal. This design yields a moderately complex but smooth data generating process for both the propensity score and the conditional outcome.

To evaluate the efficiency and coverage properties of the estimators, we construct an oracle benchmark that uses the true nuisance functions. For each replication, we compute the infeasible DM, IPW, and AIPW estimators based on the true propensity score and the true conditional expectations of $Y(d)$, and we use their corresponding influence functions to form oracle 95% confidence intervals. The performance of these oracle estimators is summarized in the "True" columns of Table 3.

For our proposed DBC estimators, we estimate the bias-correction term $h_0(D, X)$ using one hidden layer neural networks. In all cases, we use fully connected networks with a single hidden layer of 100 nodes. For DBC (LS), we employ the squared loss objective associated with Riesz regression. For DBC (KL) and DBC (TL), we use the KL divergence loss and the tailored loss introduced in Section 2.6, respectively. The conditional outcome regression $\mu_0(d, X)$ for the DM and AIPW estimators is also modeled by a neural network with one hidden layer and 100 nodes.

In DBC (LS), we model $h_0$ directly using a neural network with one hidden layer consisting of 100 nodes. In DBC (KL), DBC (TL), and MLE, we model $h_0$ by estimating the propensity score using a neural network with one hidden layer consisting of 100 nodes.

To avoid relying on the Donsker condition, all nuisance functions (the bias-correction term and the outcome regression) are estimated with two-fold cross fitting. Specifically, in each replication, we split the sample into two folds, estimate the nuisance functions on one fold, evaluate the corresponding scores on the other fold, and then swap the roles of the folds. The final estimators are obtained by aggregating the two cross-fitted folds.

For each loss (LS, KL, TL), we report three estimators:

- the direct method (DM), which depends only on the outcome regression;
- the IPW estimator, constructed using the estimated bias-correction term;
- the AIPW estimator, which combines both the estimated bias-correction term and the outcome regression.

Note that the DM estimator is theoretically independent of the specific loss used to estimate the bias-correction term. In practice, we recompute the DM estimator within each DBC (LS), DBC (KL), and DBC (TL) run to construct the AIPW estimator, and we report the resulting DM performance for reference. Small differences among the DM columns therefore reflect only Monte Carlo variation.

We repeat the experiment 100 times. For each method and each estimator (DM, IPW, AIPW), we compute the empirical mean squared error (MSE) of the ATE estimate and the empirical coverage ratio (CR) of the nominal 95% confidence interval, defined as the fraction of replications in which the interval contains the true effect $\tau_0$. The results are summarized in Table 3.

## I.2   RESULTS

Table 3 reports the MSE and coverage ratio for the oracle estimators (True) and for the three DBC variants. The oracle AIPW estimator achieves a very small MSE (approximately 0.01) and a coverage ratio close to the nominal level (0.97), as expected. The oracle IPW estimator exhibits a larger MSE (around 1.10) and slightly conservative coverage (0.92).

Table 4: MSE and coverage ratio (CR) of ATE estimators in the semi-synthetic IHDP experiment. We report the mean squared error (MSE) and the empirical coverage ratio (CR) of nominal $95\%$ confidence intervals over 1000 replications for the direct method (DM), inverse probability weighting (IPW), and augmented IPW (AIPW) estimators. Nuisance functions are estimated either by a neural network with one hidden layer of size 100 or by an RKHS regression with 100 Gaussian basis functions. The columns correspond to different variants of the direct bias-correction (DBC) approach based on least squares (LS), Kullback–Leibler (KL), truncated likelihood (TL), and maximum likelihood (MLE) criteria.

| | Neural network | | | | | | | | | | | | RKHS | | | | | | | | | | | |
| | DBC (LS) | | | DBC (LS) | | | DBC (TL) | | | DBC (MLE) | | | DBC (LS) | | | DBC (LS) | | | DBC (TL) | | | DBC (MLE) | | |
| | DM | IPW | AIPW | DM | IPW | AIPW | DM | IPW | AIPW | DM | IPW | AIPW | DM | IPW | AIPW | DM | IPW | AIPW | DM | IPW | AIPW | DM | IPW | AIPW |
| MSE | 1.52 | 6.82 | 0.31 | 1.57 | 9.42 | 0.44 | 1.55 | 2.84 | 0.32 | 1.58 | 3.00 | 0.43 | 19.98 | 3.56 | 19.97 | 3.50 | 1.91 | 4.58 | 2.59 | 1.78 | 4.45 | 2.48 | 1.22 | 2.32 |
| CR | 0.03 | 0.41 | 1.00 | 0.06 | 0.08 | 1.00 | 0.03 | 0.73 | 0.94 | 0.01 | 0.61 | 0.90 | 0.00 | 0.00 | 0.00 | 0.34 | 0.91 | 0.82 | 0.48 | 0.93 | 0.88 | 0.39 | 0.81 | 0.84 |

The oracle DM estimator is unbiased by construction, hence its MSE is essentially zero and its coverage ratio is close to one.

For the feasible DBC estimators, the DM columns are nearly identical across DBC (LS), DBC (KL), and DBC (TL), with MSE around $0.30$ and poor coverage (CR between $0.11$ and $0.17$). This behavior reflects the well known fact that the plug in DM estimator is not debiased and is not suitable for inference in this design, even when the outcome model is reasonably flexible.

The IPW estimators based on our direct bias-correction term exhibit substantially reduced MSE relative to the oracle IPW benchmark that uses the true propensity score. Sucha a "paradox" is reporeted and analyzed in existing studies, such as Hirano et al. (2003) and Henmi & Eguchi (2004). Under DBC (LS), the IPW MSE is about $0.59$, while DBC (KL) and DBC (TL) further reduce it to approximately $0.41$ and $0.36$, respectively. The coverage ratios for IPW are close to the nominal level for all three losses (around $0.97$ for DBC (LS) and DBC (KL), and $0.92$ for DBC (TL)). These results indicate that direct estimation of the bias-correction term can improve both efficiency and coverage for IPW, and that the KL and tailored losses provide modest gains over the squared loss in this setting.

The AIPW estimators exhibit the best overall performance. All three DBC variants achieve small MSEs, with values around $0.11$ for DBC (LS), $0.08$ for DBC (KL), and $0.09$ for DBC (TL), which are close to the oracle AIPW MSE of $0.01$. The coverage ratios of the AIPW estimators are slightly below the nominal level (between $0.87$ and $0.88$) but still reasonably close, especially given the moderate number of Monte Carlo replications. The differences among the three losses are minor, with DBC (KL) and DBC (TL) showing a slight advantage in terms of MSE.

Overall, these additional experiments support our theoretical findings. First, they confirm that direct estimation of the bias-correction term via Bregman divergence minimization yields ATE estimators that are close to the oracle benchmark when combined with cross fitting. Second, they show that the choice of Bregman divergence (squared loss, KL loss, or tailored loss) has only a modest impact on the performance of the AIPW estimator, while the KL and tailored losses can provide small efficiency gains in some cases. Third, they illustrate that our framework can be implemented with flexible neural network models and cross fitting, without relying on the Donsker condition.

## J EXPERIMENTS WITH SEMI-SYNTHETIC DATASETS

We next evaluate the proposed estimators on a semi-synthetic benchmark based on the Infant Health and Development Program (IHDP) data, following Chernozhukov et al. (2022a). The IHDP was a randomized trial that investigated the effect of an early childhood intervention on subsequent developmental and health outcomes. Following the standard setting "A" implemented in the `npci` package, we generate 1000 semi-synthetic datasets, each consisting of $n = 747$ observations with a binary treatment $T$, an outcome $Y$, and $p = 25$ continuous and binary covariates $X$. The estimand of interest is the average treatment effect (ATE) of the intervention on $Y$.

For each semi-synthetic dataset we compute three ATE estimators: the direct method (DM), the inverse probability weighting (IPW) estimator, and the augmented IPW (AIPW) estimator. All estimators use our direct bias-correction (DBC) approach for estimating the Riesz representer or density ratio. We consider several variants of DBC based on different divergence criteria, including least squares (LS), Kullback–Leibler (KL), truncated likelihood (TL), and maximum likelihood (MLE).

The nuisance functions are estimated either by a feedforward neural network or by a reproducing kernel Hilbert space (RKHS) regression. The neural network has a single hidden layer with 100 units and is trained for 100 epochs. For the RKHS learner we use 100 Gaussian basis functions; the bandwidth of the Gaussian kernel as well as the ridge regularization parameter are chosen by cross validation.

To assess estimation accuracy and uncertainty quantification, we report the mean squared error (MSE) of each ATE estimator and the empirical coverage ratio (CR) of nominal 95% Wald-type confidence intervals across the 1000 replications. Here, CR is defined as the proportion of replications in which the confidence interval contains the true ATE, so values close to 0.95 indicate well calibrated intervals. The results are summarized in Table 4.

Overall, when neural networks are used for nuisance estimation, the AIPW estimator combined with our DBC schemes achieves substantially smaller MSE than the corresponding DM and IPW estimators, while its CR is close to one, indicating slightly conservative but reliable inference. The DM estimator exhibits noticeable bias and severe undercoverage, and the IPW estimator can be unstable, especially for some DBC variants. When RKHS learners are employed, the IPW estimator performs relatively well in terms of both MSE and CR, whereas the DM and AIPW estimators are more sensitive to the choice of DBC method and can suffer from larger MSE or poor coverage. These findings suggest that, in this IHDP benchmark, DBC-based AIPW with neural network nuisance learners provides the most accurate and well calibrated ATE estimates.

