# OpenReview forum: "Direct Bias-Correction Term Estimation for Average Treatment Effect Estimation"
_ICLR.cc/2026/Conference — Submitted to ICLR 2026_

### Official Review · Reviewer_ZxhR · 2025-10-25

**Soundness:** 3
**Presentation:** 3
**Contribution:** 1
**Rating:** 2
**Confidence:** 5

**Summary:**

This work presents a way to estimate the Riesz representer of a well-studied quantity, the average treatment effect (ATE). In Section 1.2, the authors claim the following contributions:

1. The proposal of a framework for direct bias-correction term estimation.
2. The theoretical analysis of the estimator obtained via direct bias-correction term estimation.
3. Generalizing this framework from L2 losses to Bregman losses.
4. Unifying the existing literatures between Riesz regression and covariate balancing.

I respectfully disagree with the authors that #1, #2, and #4 are novel contributions. See my answer in **Weaknesses** for more detail.

**Strengths:**

To the best of my knowledge, contribution #3 is novel.

**Weaknesses:**

As the authors acknowledge later (beginning of Section 5.1), the framework in #1 is equivalent to Riesz regression, an existing framework that has already been used to estimate ATEs and linear regression functionals more generally. As a result, the extensive theoretical analyses of Riesz regression already apply to this method and the resulting estimator, e.g., from these works:

https://arxiv.org/pdf/2104.14737
https://arxiv.org/pdf/1809.05224
https://arxiv.org/pdf/2110.03031

#4 is discussed in this paper (e.g., Section 2.2):

https://arxiv.org/pdf/2304.14545

Finally, while #3 may be novel, the authors don't make a compelling case for why it's worth optimizing a Bregman divergence instead of just an L2 divergence. Making such a case is important since, when Cauchy-Schwarz is applied to the von Mises expansion, the L2 distance between the estimated and true Riesz representer emerges naturally in an upper bound. Hence, clearly making the L2 distance small will also reduce the bias. A clear case would be needed to explain why some other Bregman divergence would be preferred. While making such contributions would be an interesting direction for improvement, I'm skeptical that the improvement to the manuscript would be sufficient to outweigh my other novelty concerns.

**Questions:**

See above.

---

> ### Author Response · Authors · 2025-11-17
> **Reply to Reviewer ZxhR**
>
> We appreciate for your detailed feedback. Reflecting your comments, we have updated the draft. For the details of the updated contents, please see the comment titled "For all reviewers". We answer your questions below.
>
>
> **Q1.** As the authors acknowledge later (beginning of Section 5.1), the framework in #1 is equivalent to Riesz regression, an existing framework that has already been used to estimate ATEs and linear regression functionals more generally. As a result, the extensive theoretical analyses of Riesz regression already apply to this method and the resulting estimator.
>
> **A1.**
> - This point is also our contribution. That is, Riesz regression is almost identical to Least-Squares Importance Fitting (LSIF) in density ratio estimation (Kanamori et al., 2009).
> - Note that we do not deny the novelty of Riesz regression, since it is general in the sense that it can be applied to various estimation problems related to the Riesz representer.
> - This finding is important in the sense that we can deepen the understanding of the methods and prevent reinvention and methodological confusion.
>      - For example, in the recent study about covariate shift adaptation by Chernozhukov et al., they propose Riesz regression for that problem, but the method is the same as covariate shift adaptation with LSIF proposed in Kanamori et al. (2009), except for the regression adjustment term for variance reduction.
>      - Convergence rate analysis of Riesz regression is provided by Chernozhukov et al. (2021), but we can find other general results in the DRE literature, which can be directly used for Riesz regression.
>      - It is known that in DRE (that is, Riesz representer estimation), overfitting is easily occurred due to the form of the objective function. To mitigate the overfitting, there have been proposed various methods such as telescoping density ratio (Rhodes et al., 2020) and nonnegative correction (Kato and Teshima, 2022).
> - We can show that nearest neighbor matching is a special case of Riesz regression and LSIF.
>
> ------
>
> **Q2.** #4 is discussed in this paper (e.g., Section 2.2). (Bruns-Smith et al., 2025)
>
> **A2.** After the submission to ICLR, we have reconsidered our contributions and found that Bregman divergence minimization and covariate balancing have a close connection. We explain this in the subsequent A3. We also added a new section "3 AUTOMATIC COVARIATE BALANCING" in the revised draft.
>
> ------
>
> **Q3.** Finally, while #3 may be novel, the authors don't make a compelling case for why it's worth optimizing a Bregman divergence instead of just an L2 divergence. Making such a case is important since, when Cauchy-Schwarz is applied to the von Mises expansion, the L2 distance between the estimated and true Riesz representer emerges naturally in an upper bound. Hence, clearly making the L2 distance small will also reduce the bias. A clear case would be needed to explain why some other Bregman divergence would be preferred. While making such contributions would be an interesting direction for improvement, I'm skeptical that the improvement to the manuscript would be sufficient to outweigh my other novelty concerns.
>
> **A3.**
> - The generalization as Bregman divergence minimization is meaningful for model specification of the bias-correction term (Riesz representer).
> - Recall that Bregman divergence yields various loss functions by changing $g$. For example, we can derive the squared loss and KL divergence. Bregman divergence minimization with squared loss $g$ coincides with Least-Squares Importance Fitting (LSIF) in density-ratio estimation (DRE) (Kanamori et al., 2009) and Riesz regression in debiased machine learning (Chernozhukov et al., 2021). Bregman divergence minimization with KL divergence $g$ coincides with KLIEP in DRE (Sugiyama et al., 2008) and tailored loss minimization in covariate balancing (Zhao, 2019).
> - We find that under specific choices of models and losses, the bias-correction term estimated via Bregman divergence minimization automatically has the covariate balancing property.
> - For example, if we use linear models for $h_0$ and train it by squared loss minimization (equivalently, LSIF and Riesz regression), the dual problem becomes the stable balancing weights by Zubizarreta (2015).
> - If we model $h_0$ via a propensity score modeled by a logistic model and train it by KL divergence minimization (equivalently, KLIEP and tailored loss), the dual problem becomes the entropy balancing weights by Hainmueller (2012).
> - In general, covariate balancing is recognized as a desirable property. Surprisingly, under specific choices of models and loss functions, we can automatically ensure this property without solving the covariate balancing problem explicitly.

---

> ### Author Response · Authors · 2025-11-17
> **Reply to Reviewer ZxhR (cont.)**
>
> **Final remark.**
> Our contribution is the generalization of existing methods such as Riesz regression, covariate balancing weights, and density-ratio estimation. This generalization allows us to (i) import existing theoretical results while preventing reinvention, (ii) obtain insight into how to model the bias-correction term.
>
> We emphasize that at least in ATE and ATT estimation, Riesz regression (Chernozhukov et al., 2021) is almost identical to LSIF and the tailored loss for balancing weights (Zhao, 2019) is almost identical to KLIEP. In addition, their dual problems are stable balancing weights and entropy balancing weights, respectively, under specific model choices (These results are shown in Bruns-Smith et al., 2025 and Zhao, 2019). Our framework unites them under a single existing density ratio estimation framework, which also allows us to use theoretical and methodological results in the density ratio estimation literature.
>
> For this literature, we are concerned that there exist various reinventions or parallel inventions. We believe that this is because existing studies refer to their methods with their own terminologies and they do not communicate with other communities. Our result resolves this unfortunate situation and boosts the research of this literature by bridging them under a single framework.
>
> -------
>
> **Revision.**
> - We revised the draft so that our framework unites the existing literature.
> - In the next revision, we will add more experimental studies to support our claims.
>
> -------
>
> **Refences:**
> - David Bruns-Smith, Oliver Dukes, Avi Feller, and Elizabeth L Ogburn. Augmented balancing weights as linear regression. Journal of the Royal Statistical Society Series B: Statistical Methodology, 2025.
> - Qingyuan Zhao. Covariate balancing propensity score by tailored loss functions. The Annals of Statistics, 47(2):965–993, 2019.
> - Jens Hainmueller. Entropy balancing for causal effects: A multivariate reweighting method to produce balanced samples in observational studies. Political Analysis, 20(1):25–46, 2012.
> - Victor Chernozhukov, Michael Newey, Whitney K Newey, Rahul Singh, and Vasilis Srygkanis. Automatic debiased machine learning for covariate shifts, 2025. arXiv: 2307.04527.
> - Takafumi Kanamori, Shohei Hido, and Masashi Sugiyama. A least-squares approach to direct importance estimation. Journal of Machine Learning Research, 10(Jul.):1391–1445, 2009.
> - Victor Chernozhukov, Whitney K. Newey, Victor Quintas-Martinez, and Vasilis Syrgkanis. Automatic debiased machine learning via Riesz regression, 2021. arXiv:2104.14737.
> - Masashi Sugiyama, Taiji Suzuki, Shinichi Nakajima, Hisashi Kashima, Paul von Bunau, and Motoaki Kawanabe. Direct importance estimation for covariate shift adaptation. Annals of the Institute of Statistical Mathematics, 60(4):699–746, 2008.
> - Jens Hainmueller. Entropy balancing for causal effects: A multivariate reweighting method to produce balanced samples in observational studies. Political Analysis, 20(1):25–46, 2012.
> - Masahiro Kato and Takeshi Teshima. Non-negative bregman divergence minimization for deep direct density ratio estimation. In International Conference on Machine Learning (ICML), 2021.
> - Benjamin Rhodes, Kai Xu, and Michael U. Gutmann. Telescoping density-ratio estimation. In Advances in Neural Information Processing Systems (NeurIPS), 2020.

---

> > ### Comment · Reviewer_ZxhR · 2025-11-25
> >
> > Thank you for your replies. It sounds that you essentially agree that #1, #2, and #4 from my original review are not novel methodological contributions, but are better viewed as clarifying and unifying connections between existing methods. This primarily positions the paper as a conceptual unification of the existing literature rather than as introducing substantially new methodology or theory beyond what is already available in existing work.
> >
> > While the contribution #3 does indeed remain novel, the authors have not made a compelling case that this novelty yields a meaningful improvement over the special cases already studied in detail in the literature.
> >
> > It's also worth emphasizing that - as you've noted - your unification only applies in the ATE case. Riesz regression is more general - applying to any bounded linear regression functional - and the Riesz representers of those functionals are not always density ratios.
> >
> > For those reasons, I will maintain my score.

---

> ### Author Response · Authors · 2025-11-25
> **Reply to Reviewer ZxhR**
>
> We thank the reviewer for the careful rereading of our paper and for the detailed comments. We emphasize the following points to clarify our contributions.
>
> - Importantly, our method can be applied to more general problems beyond density-ratio related problems, such as ATE and ATT estimation, by utilizing the idea of Riesz regression. We discovered this extension after submitting the present paper to ICLR, and it is currently available on a Preprint server. In that work, we use the Riesz regression framework with the Bregman divergence, and show that essentially all applications treated in Riesz regression can be formulated as the Bregman divergence minimization problem. If necessary, and with due consideration for anonymity, we can either cite that paper or incorporate its main results into the current manuscript, while consulting the Area Chair for their opinion.
>
> - We believe that organizing and unifying existing work is important. For example, at least for ATE and ATT estimation, Riesz regression of Chernozhukov et al. (2021) is, under suitable modeling assumptions if needed, mathematically equivalent to LSIF (Kanamori et al., 2009), stable balancing weights (Zubizarreta et al., 2015), and kernel-based covariate functional balancing (Wong et al., 2015). It is undesirable that essentially the same methods are repeatedly reinvented across different research areas.
>
> - Moreover, within our framework we can show that nearest neighbor matching is also mathematically equivalent to Riesz regression and LSIF. In a recent paper, Lin et al. (2023) link nearest neighbor matching and density ratio estimation and present this as a new method for density ratio estimation, but we find that their estimator is in fact isomorphic to LSIF and Riesz regression. We also discovered this connection after the original ICLR submission and wrote it up as a separate preprint. Because we were concerned that citing the preprint might break anonymity, we instead incorporated the corresponding arguments into the appendix of the current anonymous manuscript.
>
>
> -------
>
> - Zhexiao Lin, Peng Ding, and Fang Han. Estimation based on nearest neighbor matching: from density ratio to average treatment effect. Econometrica, 91(6):2187–2217, 2023.

---

> > ### Comment · Reviewer_ZxhR · 2025-11-25
> >
> > Per our earlier messages in this conversation, many of these equivalences have already been shown in the literature:
> >
> > - Bruns-Smith et al. (2025) showed an equivalence between instances of stable balancing weights and Riesz regression
> > - Zhao (2019) already showed an equivalence between instances of entropy balancing weights and Riesz regression
> > - Lin et al. (2023) already make a connection between nearest neighbor matching and density ratio estimation
> >
> > Thus, in my view, these works already address, to a substantial extent, the concern you raise that “essentially the same methods are repeatedly reinvented across different research areas.”

---

> ### Author Response · Authors · 2025-11-25
>
> We sincerely appreciate your quick reply. Regarding the points you raised, we would like to offer the following clarifications:
>
> > Bruns-Smith et al. (2025) showed an equivalence between instances of stable balancing weights and Riesz regression.
> > Zhao (2019) already showed an equivalence between instances of entropy balancing weights and Riesz regression.
>
> - We further find that Riesz regression (as discussed in Chernozhukov et al. and Bruns-Smith et al.) and the tailored loss function proposed by Zhao (2019) can both be understood within a single framework, Bregman divergence minimization, with different choices of loss functions. The former adopts squared loss in the Bregman divergence, while the latter utilizes KL divergence.
>
> - While Bruns-Smith et al. (2025) and Zhao (2019) point out the equivalence, the connection between covariate balancing (via the dual formulation) and density ratio estimation methods had already been partially pointed out in earlier work, such as Sugiyama et al. (2011). Furthermore, a related duality in the context of density estimation was discussed even earlier by Silverman (1982). We believe it is valuable to connect these strands of the literature.
>
> > Lin et al. (2023) already make a connection between nearest neighbor matching and density ratio estimation.
>
> - Lin et al. claims that they developed a novel density ratio estimation method and linked nearest neighbor matching to density ratio estimation. The finding between the density ratio and nearest neighbor matching is indeed novel. On the other hand, in our study and our subsequent work, we also reconsider the novelty of their density ratio method. By reformulating their method, we find that it falls within the framework of LSIF and Riesz regression. In this sense, there is another instance of equivalence, which has not been pointed out.
>
> -------
>
> References:
> - Masashi Sugiyama, Taiji Suzuki, and Takafumi Kanamori. Density ratio matching under the Bregman divergence: A unified framework of density ratio estimation. Annals of the Institute of Statistical Mathematics, 64, 2011.
> - B. W. Silverman. On the estimation of a probability density function by the maximum penalized likelihood method. The Annals of Statistics, 10(3):795--810, 1982.

---

### Official Review · Reviewer_RVai · 2025-10-28

**Soundness:** 3
**Presentation:** 3
**Contribution:** 2
**Rating:** 6
**Confidence:** 3

**Summary:**

This paper proposes a novel approach to estimating the Average Treatment Effect (ATE) by directly estimating the bias-correction term $h_0$, based on the key insight that the objective function can be reformulated without requiring prior knowledge of the true $h_0$​. The authors first present a basic least squares formulation and then extend the method using Bregman divergence minimization. This general framework unifies and broadens existing approaches such as Riesz regression and covariate balancing.

**Strengths:**

This paper proposes a general framework for directly minimizing the error between the true bias-correction term $h_0$​ and its estimate $\hat h$, and establishes connections to several existing estimators, including Riesz regression and covariate balancing methods.

**Weaknesses:**

1. Although the paper emphasizes that the goal is not to estimate the function $r$ directly, the theoretical analysis still appears to rely on its convergence. In particular, the root-n convergence of the ATE estimator is established through the joint convergence of both $r$ and the outcome model $\mu$, following the standard ATE analysis framework.  Since $h_0$ is directly estimated, I am wondering if there are theoretical advantages that can be demonstrated? For example, in the covariate balancing paper "Kernel-based Covariate Functional Balancing for Observational Studies", it can be shown that root-n convergence for ATE is achieved without any modeling assumptions on the weights and without requiring estimation of the regression function.

2. For the comparison with other ATE estimators, it would be beneficial to include a broader set of benchmarks, such as augmented covariate balancing estimators and various weighted estimators commonly used for ATE. Additionally, presenting inference results—such as coverage probabilities of confidence intervals—would strengthen the empirical evaluation. Finally, incorporating a real data application would provide valuable insight into the practical performance and robustness of the proposed estimator.

**Questions:**

See above.

**Details Of Ethics Concerns:**

N/A.

---

> ### Author Response · Authors · 2025-11-22
> **Reply to Reviewer RVai**
>
> Thank you for your feedback. Reflecting your comments, we have updated the draft. For the details of the updated contents, please see the comment titled "For all reviewers". We answer your questions below.
>
> -------
> **Q1.1.** Although the paper emphasizes that the goal is not to estimate the function $r$ directly, the theoretical analysis still appears to rely on its convergence. In particular, the root-n convergence of the ATE estimator is established through the joint convergence of both $r$ and the outcome model $\mu$, following the standard ATE analysis framework.
>
> **A1.** We show the root-n convergence of the ATE estimator using the joint convergence of both $r$ and the outcome model $\mu$ in order to retain generality. If we restrict models to specific classes and add additional assumptions, we can relax these conditions. However, at the model selection level, our form of theoretical results is known to be among the most general (Chernozhukov et al., 2018; Schuler and van der Laan, 2024). In this study, we provide a general result that allows us to use various models for estimating $h_0$ or $r_0$.
>
> -------
>
> **Q1.2.** Since $r$ is directly estimated, I am wondering if there are theoretical advantages that can be demonstrated? For example, in the covariate balancing paper "Kernel-based Covariate Functional Balancing for Observational Studies", it can be shown that root-n convergence for ATE is achieved without any modeling assumptions on the weights and without requiring estimation of the regression function.
>
> **A2.** Firstly, our framework also includes "Kernel-based Covariate Functional Balancing for Observational Studies" (Wong et al., 2017) as a special case, although the theoretical analysis is quite different. When using the squared loss and RKHS for modeling $h_0$, we can obtain the objective function of Wong et al. (2017). This is also pointed out in Bruns-Smith et al. (2025).
>
> We conjecture that Wong et al. (2017) also implicitly uses modeling assumptions on the weights. We agree that the results in Wong et al. (2017) are surprising, but we believe that the relationship among the assumptions needs to be carefully assessed. Efficient ATE estimation without estimating the regression function is classically studies in Hirano et al. (2003). However, both Hirano et al. (2003) and Wong et al. (2017) require some restrictions. Investigating mechanisms of their results is an open problem that we plan to discuss in the next revision or in future work.
>
> In Wong et al. (2017), the authors consider covariate balancing on basis functions belonging to an RKHS $H$ with finite entropy, and they assume that the regression function belongs to the RKHS $H$. Although they do not impose specific assumptions on the propensity scores, these assumptions implicitly induce modeling restrictions on the propensity scores.
>
> The objective of Wong et al. (2017) can be written as squared loss minimization between $h_0$ and $h$ belonging to $H$. However, $h_0$ is not assumed to be in $H$, while the regression function is assumed to belong to $H$. This result is interesting, and we will continue investing the point.
>
> In contrast, the entropy of $H$ is assumed to be finite, that is, the RKHS belongs to the Donsker class. Our result does not require the Donsker condition; if it is violated, we can still show root-n convergence by using cross-fitting and suitable joint convergence rate conditions for $h_0$ and $\mu_0$.
>
>
> -------
>
> **Q3.**For the comparison with other ATE estimators, it would be beneficial to include a broader set of benchmarks, such as augmented covariate balancing estimators and various weighted estimators commonly used for ATE. Additionally, presenting inference results—such as coverage probabilities of confidence intervals—would strengthen the empirical evaluation. Finally, incorporating a real data application would provide valuable insight into the practical performance and robustness of the proposed estimator.
>
> **A3.** We have added experimental results using additional simulation datasets and the IHDP dataset. In these experiments, we report coverage probabilities of confidence intervals.
>
> -------
>
> **Reference**
> - Keisuke Hirano, Guido Imbens, and Geert Ridder. Efficient estimation of average treatment effects using the estimated propensity score. Econometrica, 2003.
> - Raymond K W Wong and Kwun Chuen Gary Chan. Kernel-based covariate functional balancing for observational studies, 2016.
> - David Bruns-Smith, Oliver Dukes, Avi Feller, and Elizabeth L Ogburn. Augmented balancing weights as linear regression. 2025.
> - Alejandro Schuler and Mark van der Laan. Introduction to modern causal inference. 2024.
> - Victor Chernozhukov, Denis Chetverikov, Mert Demirer, Esther Duflo, Christian Hansen, Whitney Newey, and James Robins. Double/debiased machine learning for treatment and structural parameters. 2018.

---

> > ### Comment · Reviewer_RVai · 2025-11-26
> >
> > I appreciate the authors’ response. While other reviewers have noted substantial overlap between this work and existing literature, I tend to view this paper as a more general framework that encompasses many prior works as special cases. However, I do not find additional significant theoretical contributions that offer new insights beyond what is already established. Therefore, I will maintain my current rating for this paper.

---

> ### Author Response · Authors · 2025-12-03
>
> Thank you very much for the positive and encouraging feedback.
> - Regarding the theoretical contribution, we believe that providing a general framework that unifies many existing results is already a substantial contribution in itself. In addition, the paper conducts a detailed estimation error analysis and establishes minimax optimality.
> - Building on this work, we have developed subsequent research that further generalizes Riesz regression with Bregman divergences beyond ATE estimation, and that, within a density ratio estimation framework, shows that nearest-neighbor matching can be viewed as a special case of Riesz regression. These projects challenge some of the claims made in papers published in leading machine learning conferences and leading economics journals, and we believe their contributions are significant. However, we consider it more appropriate to present those results in separate papers, for example as comment papers submitted to journals that the original papers are published, rather than incorporating them into the current submission. Moreover, in order to avoid any potential violation of the double-blind review policy, we are not in a position to cite these follow-up works directly at this stage. If the program committee finds it helpful, we would be happy to discuss with the Area Chair how best to handle these related results within the constraints of the anonymity policy.
>
> We will continue to work on improving the paper.

---

### Official Review · Reviewer_E3JL · 2025-10-30

**Soundness:** 3
**Presentation:** 2
**Contribution:** 2
**Rating:** 2
**Confidence:** 4

**Summary:**

This paper addresses the estimation of the average treatment effect (ATE) in observational studies. Instead of using the standard approach which estimates the propensity score and then inverts it, the authors propose to estimate the inverse of the propensity score directly. They develop an estimator under this framework and analyze its theoretical properties.

**Strengths:**

The paper tackles an important and classical problem in causal inference -- ATE estimation under confounding.

The authors provide a theoretical analysis of their proposed estimator, which adds rigor to their contribution.

**Weaknesses:**

The core idea appears to overlap substantially with the extensive literature on balancing weights, which already focuses on constructing weights that directly achieve covariate balance without explicitly estimating the propensity score. Relevant prior works include Imai & Ratkovic (2014), Zubizarreta (2015), Chan et al. (2016), Zhao & Percival (2016), Fan et al. (2016), Wong & Chan (2018), Zhao (2019), and Wang & Zubizarreta (2020). The paper does not clearly distinguish itself from these studies or state what is fundamentally new.

Estimating the inverse propensity score remains an intermediate step toward estimating the treatment effect. It is not evident what benefit is gained from focusing on this quantity rather than directly estimating the treatment effect.

The authors do not adequately discuss how their method compares in performance or robustness to existing approaches, particularly under nonsmooth or misspecified settings, where other methods (e.g., Robins et al., 2008, 2009, 2017; Yu & Wang, 2024) have explored efficiency improvements.

**Questions:**

How does the proposed method differ conceptually and practically from balancing-weight approaches such as entropy balancing or covariate balancing propensity scores?

What is the main benefit (either theoretical or empirical) of estimating the inverse of the propensity score directly rather than estimating the treatment effect directly?

---

> ### Author Response · Authors · 2025-11-17
> **Reply to Reviewer E3JL**
>
> Thank you for your feedback. Reflecting your comments, we have updated the draft. For the details of the updated contents, please see the comment titled "For all reviewers". We answer your questions below.
>
> -----
>
> **Q1**. How does the proposed method differ conceptually and practically from balancing-weight approaches such as entropy balancing or covariate balancing propensity scores?
>
> **A1.**
> - Our framework generalizes existing methods and includes the works the reviewer raised as special cases.
> - Bregman divergence yields various loss functions by changing $g$. For example, we can derive the squared loss and KL divergence. Bregman divergence minimization with squared loss $g$ coincides with Least-Squares Importance Fitting (LSIF) in density ratio estimation (DRE) (Kanamori et al., 2009) and Riesz regression in debiased machine learning (Chernozhukov et al., 2021). Bregman divergence minimization with KL divergence $g$ coincides with KLIEP in DRE (Sugiyama et al., 2008) and tailored loss minimization in covariate balancing (Zhao, 2019).
> - Under specific choices of models and losses, the bias-correction term estimated via Bregman divergence minimization automatically has the covariate balancing property.
> - For example, if we use linear models for $h_0$ and train it by squared loss minimization (equivalently, LSIF and Riesz regression), the dual problem becomes the stable balancing weights by Zubizarreta (2015).
> - If we model $h_0$ via a propensity score modeled by a logistic model and train it by KL divergence minimization (equivalently, KLIEP and tailored loss), the dual problem becomes entropy balancing weights by Hainmueller (2012).
>
> -----
>
> **Q2.** What is the main benefit (either theoretical or empirical) of estimating the inverse of the propensity score directly rather than estimating the treatment effect directly?
>
> **A2.**
> - Directly estimating the bias-correction term is beneficial for constructing efficient ATE estimators.
> - To the best of our knowledge, it is infeasible to estimate the treatment effect directly. There are several very specific exceptional cases, such as when the conditional ATE $E[Y(1) - Y(0) | X]$ is constant across $X$. However, these cases are restrictive and not usually discussed in this literature.
> - A nearly direct estimation method of ATE is to regress $Y$ on $(D, X)$ to estimate $E[Y|D, X]$, then estimate the conditional ATE as $\hat{E}[Y|1, X] - \hat{E}[Y|0, X]$, but this approach usually yields an inefficient ATE estimator, since the MSE is larger than the lower bound, that is, the efficiency bound. For such a nearly direct ATE estimator, one step bias correction using the bias-correction term has been used in various studies, such as targeted maximum likelihood estimation and double debiased machine learning (Schuler and van der Laan, 2024).
> - Thus, direct ATE estimation is usually out of interest in this literature, since it is infeasible except in unrealistic situations. The nearly direct ATE estimation mentioned above is also known to yield an inefficient estimator, and for its bias reduction, the bias-correction term plays an important role. The prior works about covariate balancing weights raised by the reviewer also take this stance. In addition, TMLE and debiased machine learning methods also follow this stance.
>
> -----
>
> **Revision.**
> - We revised the draft so that our framework unites the existing literature.
> - In the next revision, we will add more experimental studies to support our claims.
>
> -------
>
> **References:**
> - Takafumi Kanamori, Shohei Hido, and Masashi Sugiyama. A least-squares approach to direct importance estimation. Journal of Machine Learning Research, 10(Jul.):1391–1445, 2009.
> - Victor Chernozhukov, Whitney K. Newey, Victor Quintas-Martinez, and Vasilis Syrgkanis. Automatic debiased machine learning via Riesz regression, 2021. arXiv:2104.14737.
> - Masashi Sugiyama, Taiji Suzuki, Shinichi Nakajima, Hisashi Kashima, Paul von Bunau, and Motoaki Kawanabe. Direct importance estimation for covariate shift adaptation. Annals of the Institute of Statistical Mathematics, 60(4):699–746, 2008.
> - Qingyuan Zhao. Covariate balancing propensity score by tailored loss functions. The Annals of Statistics, 47(2):965–993, 2019.
> - Jens Hainmueller. Entropy balancing for causal effects: A multivariate reweighting method to produce balanced samples in observational studies. Political Analysis, 20(1):25–46, 2012.
> - Alejandro Schuler and Mark van der Laan. Introduction to modern causal inference, 2024.

---

### Official Review · Reviewer_XcMg · 2025-10-31

**Soundness:** 3
**Presentation:** 4
**Contribution:** 3
**Rating:** 6
**Confidence:** 4

**Summary:**

This paper proposes a novel framework for direct bias-correction term estimation in the context of average treatment effect (ATE) estimation. Instead of estimating the propensity score as an intermediate step, the authors directly target the bias-correction term $h_0(D, X)$, which appears in inverseprobability weighting (IPW) and doubly robust (AIPW) estimators.
The paper shows that this term can be estimated without knowing the true propensity score by minimizing an equivalent empirical risk, inspired by direct density-ratio estimation (DRE). The authors provide theoretical guarantees under linear, RKHS, and neural network settings, demonstrate asymptotic normality for AIPW estimators incorporating their method, and extend the framework using Bregman divergence minimization, unifying existing approaches such as Riesz regression and covariate balancing.
Simulation studies compare the proposed method with Logistic regression, CBPS, and RieszNet, showing comparable or superior ATE estimation accuracy in moderate-dimensional settings.

**Strengths:**

1. Conceptual originality: The direct estimation of the bias-correction term without explicit propensity score estimation is novel and appealing, consistent with the Vapnik principle.
2. Solid theoretical backing: Provides consistency and asymptotic normality results under multiple model classes (linear, RKHS, neural networks).
3. Unified framework: The generalization via Bregman divergence elegantly connects DRE, Riesz regression, and covariate balancing.
4. Empirical validation: The proposed approach achieves competitive ATE estimation accuracy relative to existing methods, confirming its practical viability.
5. Clarity and completeness: The proofs and references are comprehensive, and the relation to prior work is clearly articulated.

**Weaknesses:**

1. Limited empirical scope: Experiments are restricted to synthetic data. Evaluation on semi-synthetic or real-world causal inference datasets (e.g., IHDP, ACIC) would strengthen the empirical claims.
2. Marginal practical improvement: While theoretically elegant, the empirical gains over existing baselines such as CBPS or RieszNet are small.
3. Computational considerations: The method's scalability and hyperparameter sensitivity (e.g., regularization in RKHS or neural networks) are not discussed.
4. Clarity on assumptions: Some theoretical results assume boundedness or Donsker class conditions that may not hold for modern deep models; discussion of these limitations could be expanded.
5. Connection to efficiency: Although asymptotic efficiency is claimed, empirical efficiency gains are not clearly demonstrated.

**Questions:**

Please refer to the weaknesses above.

---

> ### Author Response · Authors · 2025-11-21
> **Reply to Reviewer XcMg**
>
> Thank you for your feedback. Reflecting your comments, we have updated the draft. For the details of the updated contents, please see the comment titled "For all reviewer". We answer your questions below.
>
> -----
>
> **Q1.** Limited empirical scope: Experiments are restricted to synthetic data. Evaluation on semi-synthetic or real-world causal inference datasets (e.g., IHDP, ACIC) would strengthen the empirical claims.
>
> **A1.** We have added experimental results using additional simulation datasets and the IHDP dataset following the setting of Chernozhukov et al. (2022). As time permits, we will add more experiments during the rebuttal period.
>
> -----
>
> **Q2.** Marginal practical improvement: While theoretically elegant, the empirical gains over existing baselines such as CBPS or RieszNet are small.
>
> **A2.** As we explained, we find that CBPS and RieszNet are special cases of our method. Our main contribution is that various existing methods, such as RieszNet, tailored loss (Zhao, 2019), density ratio estimation, entropy balancing weights, stable balancing weights, and nearest neighbor matching, can be discussed within a single framework, direct bias-correction term estimation via Bregman divergence minimization. By changing the convex function $g$ used in the Bregman divergence and the model for the bias-correction term $h_0$, we can derive these existing methods. In other words, these methods simply correspond to different loss functions in our framework.
>
> -----
>
> **Q3.** Computational considerations: The method's scalability and hyperparameter sensitivity (e.g., regularization in RKHS or neural networks) are not discussed.
>
> **A3.** Our method is a meta algorithm that includes existing methods, and scalability and hyperparameter choices depend on the underlying method. For example, as we show that Riesz regression is almost identical to least-squares importance weighting in Kanamori et al. (2008), developed for density ratio estimation, we can employ various regularization techniques for density ratio estimation, such as telescoping density ratio (Rhodes et al., 2021) and nonnegative correction in Kato and Teshima (2022). Moreover, as we added a new section in the Appendix, nearest neighbor matching is also a special case of our method, which does not include hyperparameters except for the number of neighbors.
>
> Our result unifies those exiting studies, and scalability and hyperparameter sensitivity has been discussed in those studies. The contribution is in the unification and allows us to import such various techniques developed in other areas to causal inference.
>
> -----
>
> **Q4.** Clarity on assumptions: Some theoretical results assume boundedness or Donsker class conditions that may not hold for modern deep models; discussion of these limitations could be expanded.
>
> **A4.** Our method does not require the Donsker condition. For simplicity, we present only an example of how we construct ATE estimators. Our study focuses on nuisance parameter estimation. Under suitable convergence rate conditions, if the Donsker condition is not satisfied, we can apply cross-fitting. Although we already explained this in the previous draft, we emphasize this point more clearly in the revised draft.
>
> -----
>
> **Q5.** Connection to efficiency: Although asymptotic efficiency is claimed, empirical efficiency gains are not clearly demonstrated.
>
> **A5.** As we answered in **A1,** we have added additional experimental results to support the superiority of our methods.
>
> Moreover, our results are conceptual and theoretical and aim to introduce a new framework to discuss ATE estimation methods in a unified way. Also see **A2.**, **A3.**, and **A4.** These findings connect existing methods and provide insights into which methods we should use in practice.
>
>
> - Qingyuan Zhao. Covariate balancing propensity score by tailored loss functions. The Annals of Statistics, 47(2):965–993, 2019.
> - Takafumi Kanamori, Shohei Hido, and Masashi Sugiyama. A least-squares approach to direct importance estimation. Journal of Machine Learning Research, 10(Jul.):1391–1445, 2009.
> - Jens Hainmueller. Entropy balancing for causal effects: A multivariate reweighting method to produce balanced samples in observational studies. Political Analysis, 20(1):25–46, 2012.
> - Jose R. Zubizarreta. Stable weights that balance covariates for estimation with incomplete outcome data. Journal of the American Statistical Association, 110(511):910–922, 2015.

---

### Author Response · Authors · 2025-11-17
**For all reviewers**

Thank you to all reviewers. Reflecting the comments, we updated the draft. In addition, after the submission to ICLR, we have further found a profound relationship between our direct bias-correction term estimation method and other existing methods, such as Riesz regression and covariate balancing weights. We also include these findings.

## 1. Our contributions
We restate our contents, contributions, and findings as follows:
1. We consider estimating the bias-correction term, which is also called the Riesz representer in the automatic debiased machine learning literature and the clever covariates in the targeted maximum likelihood estimation literature.
2. We propose a general framework based on Bregman divergence minimization, which depends on a convex function $g$. By specifying the function $g$, we can obtain various loss functions. For example, if we set $g$ as a squared loss, we can obtain Riesz regression. If we set $g$ as a KL divergence, we can obtain the tailored loss.
    2.1 We find that Riesz regression in Chernozhukov et al. (2021) is the same as LSIF (Kanamori et al. 2009) in density ratio estimation (DRE).
    2.2 We also find that the tailored loss in Zhao (2019) is the same as KLIEP in DRE.
3. If we use linear models for the bias-correction term and train the model via squared loss (Riesz regression), the trained bias-correction term automatically has the covariate balancing property for the basis function. If we model the bias-correction term using the propensity score with logistic models and train the model via KL divergence loss (tailored loss), the trained bias-correction term automatically has the covariate balancing property for the basis function.

- Victor Chernozhukov, Whitney K. Newey, Victor Quintas-Martinez, and Vasilis Syrgkanis. Automatic debiased machine learning via Riesz regression, 2021. arXiv:2104.14737.
- Takafumi Kanamori, Shohei Hido, and Masashi Sugiyama. A least-squares approach to direct importance estimation. Journal of Machine Learning Research, 10(Jul.):1391–1445, 2009.
- Masashi Sugiyama, Taiji Suzuki, Shinichi Nakajima, Hisashi Kashima, Paul von Bunau, and Motoaki Kawanabe. Direct importance estimation for covariate shift adaptation. Annals of the Institute of Statistical Mathematics, 60(4):699–746, 2008.
- Qingyuan Zhao. Covariate balancing propensity score by tailored loss functions. The Annals of Statistics, 47(2):965–993, 2019.



Our findings further emphasize the following points from the above points.

### 1.1. Model-loss relationship
We newly added the point (3), ``model-loss'' relationships, as our new findings. These findings have been shown separately in different literature. For example, Bruns-Smith et al. (2025) shows the duality relationship between Riesz regression and stable balancing weights. Zhao (2019) shows the duality relationship between tailored loss minimization and entropy balancing weights. We find that these results can be unified from the viewpoint of Bregman divergence minimization.

- David Bruns-Smith, Oliver Dukes, Avi Feller, and Elizabeth L Ogburn. Augmented balancing weights as linear regression. Journal of the Royal Statistical Society Series B: Statistical Methodology, 2025.

### 1.2 Riesz regression is a reinvention of LSIF for density-ratio estimation
- Although the introduction is a bit different, Riesz regression uses the same objective, derivation steps, and motivation as those in LSIF proposed in Kanamori et al. (2009) for direct density ratio estimation.
- We can find that nearest neighbor (NN) matching is a variant of Riesz regression, by extending the finding in ..., which reports that NN matching is density ratio estimation.
- We can use various results shown for LSIF, such as convergence rates and regularization techniques. For RKHS, .... For neural networks, ...

## 2. Revision
We revised the following points:
- We added the model-loss relationship section (Section 3 AUTOMATIC COVARIATE BALANCING).
- We added an explanation that our result unifies the existing literature in the sense that
    - Riesz regression and tailored loss are both special cases of Bregman divergence minimization.
    - The dual problem of squared loss based Bregman divergence minimization is the stable balancing weights, while that of KL divergence based Bregman divergence minimization is the entropy balancing weights.

In the next revision, we will add additional experimental results using semi-synthetic data.


## 3, Replies to the reviewers
We firstly send more detailed replies to Reviewer E3JL and Reviewer ZxhR. We will send more detailed replies to Reviewer RVai and Reviewer XcMg after finishing an additional experiment.

---

> ### Author Response · Authors · 2025-11-24
> **Additional experimental results**
>
> Following the reviewers' comments, we have added experimental results using additional simulation datasets and the IHDP dataset to assess the empirical performance of our proposed algorithm. Our experimental setup follows that of Chernozhukov et al. (2022). If time permits, we will include further experiments during the rebuttal period.

---

### Meta-Review · Area_Chair_cjYe · 2026-01-05

**Summary:**

The term $h$ plays an important role in constructing efficient estimator for ATE, from the perspective of semi-parametric inference. This paper aims to learn $h$ by directly minimizing the divergence between control and treated groups, quantified by the Bregman divergence.

However, after reading the overall manuscript with reviewers, my main concern aligns with the Reviewer E3JL:
> "The core idea appears to overlap substantially with the extensive literature on balancing weights, which already focuses on constructing weights that directly achieve covariate balance without explicitly estimating the propensity score. Relevant prior works include Imai & Ratkovic (2014), Zubizarreta (2015), Chan et al. (2016), Zhao & Percival (2016), Fan et al. (2016), Wong & Chan (2018), Zhao (2019), and Wang & Zubizarreta (2020). The paper does not clearly distinguish itself from these studies or state what is fundamentally new."

To be specific, authors' rebuttal did not offer strong evidence to support such unified framework to ATE estimation. I don't think that just a unified characterization of the divergence between treated and control groups via the Bregman divergence reaches the bar of ICLR conference, as the Bregman divergence itself is already proved to cover the above-mentioned divergence metrics. Therefore, I recommend the reject.

**Reviewer Concerns:**

Reviewer E3JL:
> "The core idea appears to overlap substantially with the extensive literature on balancing weights, which already focuses on constructing weights that directly achieve covariate balance without explicitly estimating the propensity score. Relevant prior works include Imai & Ratkovic (2014), Zubizarreta (2015), Chan et al. (2016), Zhao & Percival (2016), Fan et al. (2016), Wong & Chan (2018), Zhao (2019), and Wang & Zubizarreta (2020). The paper does not clearly distinguish itself from these studies or state what is fundamentally new."

This concern is very important but authors fail to alleviate it.

**Reviewer Scores:**

No reviewer will change his score after reading the brief rebuttal.

---

### Decision · Program_Chairs · 2026-01-26

Reject